# Bridging Global Intent with Local Details: A Hierarchical Representation Approach for Semantic Validation in Text-to-SQL

## Abstract

Text-to-SQL translates natural language questions into SQL statements grounded in a target database schema. Ensuring the reliability and executability of such systems requires validating generated SQL—but most existing approaches focus only on syntactic correctness, with few addressing semantic validation (detecting misalignments between questions and SQL). As a consequence, how to achieve effective semantic validation still faces two key challenges: capturing both global user intent and SQL structural details, and constructing high-quality fine-grained sub-SQL annotations. To tackle these, we introduce HeroSQL, a hierarchical SQL representation approach that integrates global intent (via Logical Plans, LPs) and local details (via Abstract Syntax Trees, ASTs). To establish better information propagation, we further employ a Nested Message Passing Neural Network (NMPNN) to capture inherent relational information in SQL and aggregate schema-guided semantics across LPs and ASTs. Additionally, to generate high-quality negative samples, we propose an AST-driven sub-SQL augmentation strategy, supporting robust optimization of fine-grained semantic inconsistencies. Extensive experiments conducted on Text-to-SQL validation benchmarks (in-domain and out-of-domain settings) demonstrate that our approach outperforms existing state-of-the-art (SOTA) methods, achieving an average 9.40% improvement of AUPRC and 12.35% of AUROC in identifying semantic inconsistencies. It excels at detecting fine-grained semantic errors, provides large language models with more granular feedback, and ultimately enhances the reliability and interpretability of data querying platforms. Our code is anonymously available at https://anonymous.4open.science/r/HeroSQL.

## 1 Introduction

Text-to-SQL (Liu et al., 2025b; Shi et al., 2024; Li et al., 2024a) is the task of translating a natural language question into an executable Structured Query Language (SQL) statement, with the translation grounded in the schema of a target relational database. The fundamental objective of Text-to-SQL is to bridge the "semantic barrier" between unstructured user intent (e.g., natural-language queries about data insights) and structured database operations (e.g., filtering, aggregation, and joins) (Liu et al., 2025a). Recent research has evolved through three major stages: the *rule-based* stage (Rajkumar & et al., 2022; Li & Jagadish, 2014; Katsogiannis-Meimarakis & Koutrika, 2021; Yu et al., 2021), the *neural network-based* stage (Xiao et al., 2016; Lin et al., 2019; Bogin et al., 2019), and the current *pretrained model-based* stage (Li & et al., 2023; Li et al., 2023a; Gu et al., 2023). More recently, the rapid emergence of Large Language Models (LLMs) (Wang et al., 2025; Cheng et al., 2024; Lian et al., 2024) has further revolutionized Text-to-SQL by providing strong contextual understanding of complex queries and SQL semantics for powerful systems.

Despite significant progress in LLM-based Text-to-SQL systems (Xie et al., 2025; Pourreza et al., 2025; Liao et al., 2025; Lee et al., 2025), these models still frequently generate semantically incorrect queries that may execute successfully but fail to faithfully capture the user's intent, as shown in Figure 1 (Jorgensen & Shepperd, 2007; Weiss et al., 2007; Liu et al., 2025c). Unlike *syntactic validation* (e.g., ensuring SQL queries are grammatically correct and executable) with sufficient error feedback, *semantic validation* often (Somov & Tutubalina, 2025;

Askari et al., 2025; Chen et al., 2023a;c; Arcadinho et al., 2022) aims to identify and correct misalignments between the user's global intent and the model-generated query structure, thereby reducing the cost of human verification and the risk of incorrect results (Jorgensen & Shepperd, 2007; Weiss et al., 2007). However, **refining semantic validator** (Chen et al., 2023c) to **bridge global user intent and local structural database schemas**, thereby **pinpointing more fine-grained misalignment feedback**, motivating several critical research challenges:

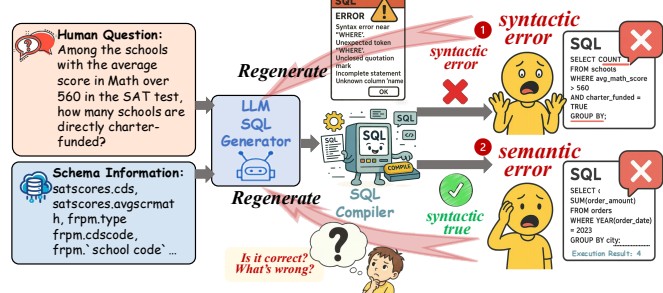

Figure 1: Syntactic errors can be easily detected and fixed, while executable SQL with semantic errors is hard to discover.

**#C1.** **Challenge in obtaining SQL representations that integrate both local structures and global intent.** Global intent captures a query's overarching purpose, encompassing high-level computational logic and data flow. Local structures, by contrast, include schema details, filter/join conditions, and predicate hierarchies that govern fine-grained execution. Integrating these two aspects into a comprehensive semantic representation remains challenging.

**#C2.** **Challenge in acquiring sufficient high-quality and fine-grained training data for Text-to-SQL validation.** Due to the inherent complexity of the Text-to-SQL task, collecting expert-annotated data that provides fine-grained feedback for iterative SQL refinement is extremely costly—especially at sub-SQL granularity. This data scarcity significantly limits validator training, preventing models from reliably distinguishing between positive and negative cases at a fine-grained level and ultimately leading to insufficient feedback quality.

To address the preceding challenges, we propose a dual-representation approach for semantic validation in Text-to-SQL. ❶ To address **#C1**, we design a **H**ierarchical **E**ncoding and **R**epresentation **O**f **SQL** queries (**HEROSQL**) for semantic validation. It utilizes the logical plan (LP) to represent the global intent of an SQL query and an abstract syntax tree (AST) to capture local structural details within each node of the logical plan. To fully capture semantic information alongside syntactic structure, we leverage a pretrained embedding model to encode input text with the necessary contextual information (e.g., database schema). We then employ a *Nested Message Passing Neural Network* (NMPNN) to aggregate schema-guided property embeddings, propagating them from local AST nodes to logical plan nodes and further to the entire query plan. ❷ To address **#C2**, we propose an **Adaptive sub-SQL Augmentation Strategy**. Specifically, we introduce semantic perturbations to the AST of gold SQL queries by modifying node attributes within the AST, which enables efficient generation of large-scale syntactically correct yet semantically incorrect negative samples. This enhances the model's ability to distinguish valid queries, particularly at the fine-grained sub-SQL level, improving robustness and generalization. Empirically, we demonstrate that this approach captures fine-grained information, effectively feeding back semantically incorrect sub-SQL for iterative generation, improving overall Text-to-SQL accuracy and significantly reducing human review costs. In summary, the main contributions of this work are as follows:

- We propose HEROSQL, a hierarchical encoding and representation approach for semantic validation in Text-to-SQL. By leveraging the hierarchical and relational information inherent in the logical plan and abstract syntax trees in SQL query, our approach is capable of capturing both global semantic and local structure information.
- We introduce an adaptive sub-SQL augmentation strategy that systematically generates challenging negative examples via AST, effectively mitigating the scarcity of annotated fine-grained SQL validation training data.
- Extensive experiments on multiple Text-to-SQL validation benchmarks demonstrate that our approach significantly outperforms existing methods in identifying semantic inconsistencies. It provides more fine-grained feedback on sub-SQL semantic errors and facilitates SQL re-optimization, thereby enhancing the reliability and interpretability of intelligent data analytics platforms.

## 2 RELATED WORK

### 2.1 SEMANTIC VALIDATION IN TEXT-TO-SQL

Traditional semantic validation methods for Text-to-SQL (Rai et al., 2023; Scholak et al., 2021; Lin et al., 2020) treat SQL queries as plain text, employing sequence-encoding models (e.g., BERT (Devlin et al., 2019), T5 (Raffel et al., 2020)) for encoding. These approaches overlook the inherent local structural information of SQL, **making it difficult to identify subtle errors within statements**. Recently, some studies have begun incorporating structural information from questions and SQL queries (Feng et al., 2020; Yu et al., 2021). Some of these methods model questions and database schemas as interconnected graphs (Wang et al., 2020; Hui & et al., 2022; Qi et al., 2022; Bazaga et al., 2023), while others represent SQL queries using abstract syntax trees (AST) (Chen et al., 2023a; Gong et al., 2025). Unlike plain text, SQL queries have a natural nested structure—their inherent hierarchical structural information is critical for representing SQL's semantic meaning. For example, SQLens (Gong et al., 2025) predicts clause-level semantic correctness in SQL queries by aggregating weak supervision signals from DB-based checks and LLM-based evaluations on AST. Yet these approaches focus largely on the syntactic details of questions and SQL queries, **making it difficult to capture global semantic intent from a macroscopic perspective**.

### 2.2 DATA AUGMENTATION FOR TEXT-TO-SQL

Due to the intrinsic complexity of the Text-to-SQL task, assembling large-scale, high-quality annotated datasets is extremely challenging. To mitigate this, numerous data augmentation strategies have been proposed to generate synthetic data, which has been proven effective in improving model generalization for Text-to-SQL (Hu et al., 2023; Tarbell et al., 2024). Early approaches primarily relied on rule-based and template-driven methods for data augmentation (Lee et al., 2025; Li et al., 2024b; Zhang et al., 2023), which **limit data diversity and domain coverage**. Recent approaches have turned to LLMs to generate questions and corresponding SQL queries (Duan et al., 2025; Li et al., 2025; Shen et al., 2023; Yang et al., 2024), enabling the creation of more diverse and domain-specific training samples. However, **the computational and financial costs associated with employing LLMs remain prohibitively high**. Notably, some methods, such as CodeBERT (Feng et al., 2020) in the code generation domain, attempt to construct negative samples using abstract syntax trees, which enables more structured and semantically meaningful examples for model training. However, **such techniques have not been explored within the Text-to-SQL domain**, presenting a promising avenue for generating large quantities of negative samples in a cost-effective manner.

## 3 PRELIMINARIES

***Task Definition 1. (Text-to-SQL Validation)*** The Text-to-SQL task aims to translate a natural language question $q$ into a corresponding SQL query $s$ that is executable on a target relational database with a predefined schema SCHEMA. As a critical post-hoc component of the Text-to-SQL pipeline, Text-to-SQL validation focuses on assessing the correctness of the generated SQL query $s$ with respect to both the input question $q$ and the database schema SCHEMA. The probability score $\hat{y}$ for the prediction can thus be formalized as Equation 1, where $f$ denotes the validation function:

$$\hat{y} = f(q, s \mid \text{SCHEMA}), \tag{1}$$

where $s$ can either be a complete SQL query or a sub-SQL fragment derived from a Logical Plan (LP). For sub-SQL fragments, we can utilize validation scores to deliver fine-grained error feedback to support LLMs in optimizing subtle semantic errors that are otherwise difficult to detect.

***Definition 2. (Intermediate Representations of SQL Queries)*** For structured queries such as SQL, it is crucial to construct *intermediate representations* (IRs) that capture both global semantics and local syntactic details. Two widely adopted IRs are the *logical plan* (LP) and the *abstract syntax tree* (AST), which provide complementary perspectives: LPs encode the high-level semantic intent of query execution, while ASTs preserve the fine-grained syntactic organization of SQL statements.

- ❶ **Logical Plan (LP).** A Logical Plan is a kind of semantic tree that offers a structured, high-level abstraction of a database query, formalizing the core sequence of operations required to derive the

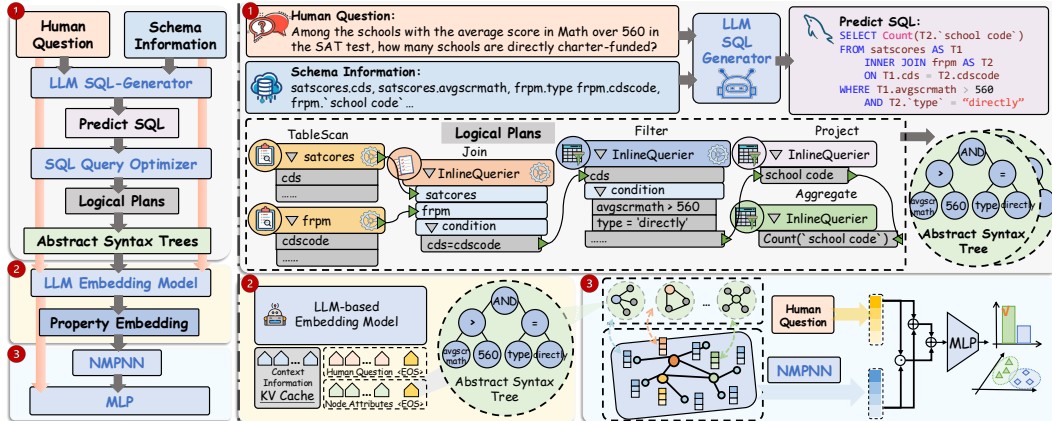

Figure 2: The overall framework of HeroSQL. Given a natural language question, database schema information, and a syntactically correct SQL query generated by an LLM, HeroSQL first applies a query optimizer to convert the query into a logical plan (LP) and parses each LP node into an abstract syntax tree (AST). The AST nodes and contextual information are then encoded into property embeddings using a pretrained LLM-based model. Finally, these embeddings are aggregated by NMPNN to form a SQL embedding, which, together with the question embedding, is fed into an MLP for semantic validation and correctness prediction.

intended result. A logical plan abstracts a query as a structured sequence of relational operations (e.g., `Filter`, `Join`, `Aggregate`) that collectively define how the query result is derived. Intuitively, an LP can be represented as a directed acyclic graph $\mathcal{G}_{\mathcal{LP}} = (V_{\mathcal{LP}}, E_{\mathcal{LP}})$, where $V_{\mathcal{LP}} = \{(o_i, a_i)\}$ includes a set of operators that each operator $o_i$ is associated with attributes $a_i$ that describe its properties, and $E_{\mathcal{LP}}$ encodes the flow of intermediate results between operators. This representation abstracts away low-level execution details while **capturing the global semantic flow of the query**.

- ❷ **Abstract Syntax Tree (AST).** In contrast, an AST captures the detailed syntactic structure of an SQL query as a hierarchical, tree-shaped form. Each node corresponds to a syntactic construct such as `SELECT`, `FROM`, `WHERE` clauses, tables, columns, or predicates. Formally, an AST is denoted as $\mathcal{AST}$ and represented as $\mathcal{G}_{\mathcal{AST}} = (V_{\mathcal{AST}}, E_{\mathcal{AST}})$, where each node $v_j^{\mathcal{A}} \in V_{\mathcal{AST}}$ contains atomic syntactic content $c_j^{\mathcal{A}}$, and edges $E_{\mathcal{AST}}$ capture the hierarchical nesting among components. By removing superficial syntax (e.g., parentheses, redundant keywords), ASTs provide a normalized view emphasizing local syntactic relationships and hierarchical structure. This granular, node-based structure **enables fine-grained analysis of local syntactic patterns and supports precise detection of subtle structural inconsistencies** in SQL statements.

## 4 METHODOLOGY

In this section, we present the methodology of HeroSQL, as illustrated in Figure 2. First, in Section 4.1, we describe how SQL queries are represented and encoded through a hierarchical structure that integrates both logical plans and abstract syntax trees. Next, in Section 4.2, we introduce our data augmentation strategies, which enrich the training corpus with diverse and fine-grained sub-SQLs for robust validation. Finally, in Section 4.3, we detail the training procedure of our SQL validation model and discuss several optimization and acceleration techniques that enhance both efficiency and scalability.

### 4.1 HIERARCHICAL ENCODING AND REPRESENTATION OF SQL QUERIES

In Section 4.1.1, we first introduce the hierarchical representation approach that effectively represents global semantics and local structure details of the SQL query with a logical plan and an abstract syntax tree. Then, we describe the embedding strategy for the text property in the context with schema information and the complete SQL query in Section 4.1.2. Finally, we present the

*Nested Message Passing Neural Network* that aggregates all information for both global semantics and local structure in the hierarchical representation of SQL queries in Section 4.1.3.

### 4.1.1 HIERARCHICAL INTERMEDIATE REPRESENTATION CONSTRUCTION

We construct a hierarchical intermediate representation (HIR) for SQL queries, which benefits from the complementary strengths of logical plans (LPs) for capturing high-level semantics and abstract syntax trees (ASTs) for modeling fine-grained structural details. This representation jointly encodes global semantic intent and local structural information.

❶ **Global Intent via Logical Plans.** To model the global intent of a SQL query, we employ query optimizers such as Apache Calcite (Begoli et al., 2018) and ORCA (Soliman et al., 2014) to convert the raw SQL query into its corresponding logical plan, represented as a directed acyclic graph: $\mathcal{G}_{\mathcal{LP}} = (V_{\mathcal{LP}}, E_{\mathcal{LP}})$, where each node $v_i^{\mathcal{LP}} = (o_i, a_i) \in V_{\mathcal{LP}}$ consists of a relational operator $o_i$ (e.g., `Filter`, `Join`, `Aggregate`) and its associated attribute $a_i$. The edges $E_{\mathcal{LP}}$ encode the flow of intermediate results between operators, capturing high-level semantic dependencies of the query.

❷ **Local Structure via Abstract Syntax Trees.** After obtaining each LP node (sub-SQL) $v_i^{\mathcal{LP}}$, the attribute $a_i$ (e.g., a filter condition or an aggregate expression) is transformed into an abstract syntax tree $\mathcal{A}_i$ to capture fine-grained syntactic structure: $\mathcal{G}_{\mathcal{A}_i} = (V_{\mathcal{A}_i}, E_{\mathcal{A}_i})$, where each node $v_j^{\mathcal{A}_i} \in V_{\mathcal{A}_i}$ contains atomic syntactic content $c_j^{\mathcal{A}_i}$, and edges $E_{\mathcal{A}_i}$ represent hierarchical composition and operator precedence.

❸ **Hierarchical Construction.** By combining the LP $\mathcal{G}_{\mathcal{LP}}$ with its node-specific ASTs $\{\mathcal{A}_i\}_{v_i \in V_{\mathcal{LP}}}$, we obtain hierarchical intermediate representation: $\mathcal{R}_{\text{SQL}} = \left( \mathcal{G}_{\mathcal{LP}}, \{\mathcal{A}_i\}_{v_i \in V_{\mathcal{LP}}} \right)$, which encodes both global semantic flow and the local syntactic details of each sub-SQL.

### 4.1.2 CONTEXT-GUIDED PROPERTY EMBEDDING FETCHING

After constructing the hierarchical intermediate representations (HIRs), we need to encode node attributes presented in text form from these HIRs. To capture semantically rich relationships between these heterogeneous attributes, we employ a pretrained LLM-based embedding model, augmented with contextual cues from the database schema and predicted SQL query. This context-guided encoding forces the model to ground its representations in the specific database schema and SQL structure, aligning the semantic spaces of question, SQL syntax, and schema elements. This alignment is critical for bridging the inherent semantic gap between user queries and SQL logical structures, particularly for schema-specific components (e.g., column names, table relationships), where naive embeddings often fail to capture domain-specific nuances.

Specifically, we enhance each input sequence $x$ by prepending a `CONTEXT` prefix, forming an augmented input $x' = [\text{CONTEXT}; x]$ where $[\cdot\,;\,\cdot]$ denotes the concatenation function. This augmented input $x'$ is fed into the embedding model $g(\cdot)$, which outputs a sequence of hidden states:

$$H = g(x'), \quad \text{with } x' = [\text{CONTEXT}; x]. \tag{2}$$

We then extract task-specific embeddings from $H$ by taking the last hidden state corresponding to the [EOS] token, yielding a condensed representation of the attribute's semantic features.

### 4.1.3 NESTED MESSAGE PASSING NEURAL NETWORK

After all node information has been encoded, we utilize a *Nested Message Passing Neural Network* (NMPNN) to encode SQL queries by using the hierarchical structure of logical plans (LPs) and their associated abstract syntax trees (ASTs). This model consists of two consecutively applied message passing neural networks: a lower-level MPNN to aggregate node information within the AST in each LP node, and a higher-level MPNN to propagate and aggregate information across LP nodes.

❶ **Lower-level Message Passing within AST.** The first stage of the NMPNN encodes the internal structure of each LP node by applying a message passing neural network over its corresponding AST.

The initial embedding $h_v^{(0)} \in H^{(0)}$ for each AST node $v \in V_{\mathcal{A}_i}$ is constructed from its syntactic type or token features. Message passing is performed for $T_{\text{AST}}$ steps according to the standard MPNN update:

$$h_v^{(t+1)} = \text{UPDATE}_{\text{AST}}\left(h_v^{(t)}, \text{AGGREGATE}_{\text{AST}}\left(\{h_u^{(t)} : u \in \mathcal{N}(v)\}\right)\right), \quad (3)$$

where $\mathcal{N}(v)$ denotes the neighbors of $v$ in the AST. After $T_{\text{AST}}$ steps, the node representations in the AST are aggregated via mean or sum pooling to produce an AST-level embedding $h_i = \text{POOLING}(\{h_v^{(T_{\text{AST}})}\}_{v \in V_{\mathcal{A}_i}})$ for logical plan node $i$.

❷ **Higher-level Message Passing over Logical Plan.** Once embeddings for all LP nodes are computed, each LP node $i \in V_{\mathcal{LP}}$ is now associated with an embedding $h_i$ derived from the corresponding AST. We apply an MPNN for $T_{\text{LP}}$ iterations over the logical plan graph as follows:

$$h_i^{(t+1)} = \text{UPDATE}_{\text{LP}}\left(h_i^{(t)}, \text{AGGREGATE}_{\text{LP}}\left(\{h_j^{(t)} : j \in \mathcal{N}_L(i)\}\right)\right), \quad (4)$$

where $\mathcal{N}_L(i)$ denotes the set of neighboring nodes of $i$ in the logical plan. After $T_{\text{LP}}$ steps, the final node embeddings are aggregated to form the representation $h_{\text{SQL}}$ for the entire SQL query, which is $h_{\text{SQL}} = \text{POOLING}(\{h_i^{(T_{\text{LP}})}\}_{i \in V_{\mathcal{LP}}})$.

By hierarchically aggregating syntactic and semantic information from both the ASTs and the LP structure, the NMPNN is able to learn expressive representations for SQL queries, which enables effective modeling of the multi-level compositionality inherent in Text-to-SQL validation tasks.

## 4.2 Data Augmentation for Text-to-SQL Validation

When training the SQL Validator, the availability of high-quality labeled data $\mathcal{D}_{\text{gold}} = \{(q_i, s_i) \mid i \in \mathbb{N}^+, y_i = 1\}$ is severely limited, and fine-grained annotations at the sub-SQL level are largely missing. Relying solely on the existing dataset is, therefore, insufficient to capture the diversity and complexity required for robust learning. To address this data scarcity issue, we introduce two complementary data augmentation strategies: ❶ **LLM-based data augmentation** (see Section 4.2.1); and ❷ **AST-driven sub-SQL augmentation strategy** (see Section 4.2.2). Together, these strategies enable us to construct a more diverse, challenging, and fine-grained training set, thereby enhancing both the robustness and generalization ability of the SQL Validator.

### 4.2.1 LLM-based Data Augmentation

LLMs exhibit remarkable generative capabilities, which can be harnessed for data augmentation. Specifically, for each question $q$ in the ground truth dataset $\mathcal{D}_{\text{gold}}$, we prompt an LLM to generate $N$ candidate SQL queries $\{\tilde{s}_i\}_{i=1}^N$ by conditioning on question $q$. Each generated SQL $\tilde{s}_i$ is then subjected to the following process:

- ❶ **Syntax Filtering**: We discard any syntactically invalid SQL queries that can be detected using an SQL compiler, and retain only those that can be parsed successfully for semantic validation.
- ❷ **Semantic Validation**: For the remaining queries, we compare the execution results of $\tilde{s}_i$ and the ground truth SQL query $s^* \in \mathcal{D}_{\text{gold}}$. A candidate query $\tilde{s}_i$ is a *valid SQL* if and only if $\text{Exec}(\tilde{s}_i) = \text{Exec}(s^*)$, and is an *invalid SQL* otherwise, where the function $\text{Exec}(\cdot)$ denotes the output of executing a SQL query on the reference database.

After filtering and validation, we get LLM-based augmented training data $\mathcal{D}_{\text{LLM}}$ both with diversified paraphrases of correct queries and with incorrect examples from semantically divergent outputs.

### 4.2.2 AST-driven Sub-SQL Augmentation Strategy

To further enlarge the training corpus and introduce challenging semantic error samples, we leverage transformations on the AST representation of valid SQL queries. Concretely, for each valid SQL $s^+$ identified in Section 4.2.1, we systematically apply a set of AST-level transformation rules $\mathcal{T}$, designed to produce semantically perturbed variants. Each transformation $T \in \mathcal{T}$ maps $s^+$ to a mutated SQL $s^- = T(s^+)$, such that the surface structure is altered while the semantic intent deviates from the original. By comparing the execution results between the augmented SQL query

and ground truth SQL query, we can identify truly semantically incorrect examples. The details of the AST-level transformation rules $\mathcal{T}$ are provided in Section D.2. Formally, the constructed semantic error sample set is:

$$\mathcal{D}_{\text{AST}} = \left\{ (q, s^-) \mid s^- = T(s^+),\ T \in \mathcal{T},\ \text{Exec}(s^-) \neq \text{Exec}(s^+) \right\}. \tag{5}$$

Beyond producing query-level semantic error samples, this augmentation also enables us to annotate *sub-SQL fragments* at the LP level. Specifically, when a transformation perturbs a certain AST node, the corresponding sub-SQL in the LP can be marked as semantically incorrect, providing fine-grained supervision signals. This design enriches the training data with more informative error patterns, allowing the model to better capture subtle inconsistencies and to generalize more effectively in distinguishing between both global and local semantic errors.

### 4.3 EFFICIENT TRAINING OF SQL VALIDATION MODEL

**Fusion of Question and SQL Embeddings.** To effectively integrate information from the natural language question and its corresponding SQL query, we use a multi-stage fusion strategy. ❶ First, given the question embedding $h_{\text{question}} \in \mathbb{R}^d$ and the SQL embedding $h_{\text{SQL}} \in \mathbb{R}^d$, we perform element-wise Hadamard multiplication (Horn, 1990; Horn & Johnson, 2012) to capture fine-grained interactions between the two modalities. The fused representation is given by $h_{\text{hadamard}} = h_{\text{question}} \odot h_{\text{SQL}}$, where $\odot$ denotes the Hadamard product. ❷ Then, we concatenate the original embeddings and the Hadamard fusion, forming $\mathbf{h} = [h_{\text{question}}; h_{\text{SQL}}; h_{\text{hadamard}}] \in \mathbb{R}^{3d}$, where $[\cdot\,;\,\cdot\,;\,\cdot]$ denotes vector concatenation. This concatenated vector aggregates both independent and interaction-based features of the input pair. ❸ Finally, the combined vector $\mathbf{x}$ is fed into a three-layer multilayer perceptron (MLP) with ReLU activation (Agarap, 2018) to produce the final output score $\hat{y}$. The process can be formally described as:

$$\hat{y} = \text{MLP}([\,h_{\text{question}}\,;\ h_{\text{SQL}}\,;\ h_{\text{question}} \odot h_{\text{SQL}}\,]). \tag{6}$$

This hierarchical fusion mechanism ensures that both shallow and deep interactions between question and SQL representations are captured, thereby facilitating effective SQL validation.

**Supervised Fine-Tuning for Text-to-SQL Validation.** The Text-to-SQL validation task is formulated as a binary classification task, where the objective is to determine if a given SQL query is correct. Specifically, our model outputs a continuous score $\hat{y} \in [0, 1]$ for each input instance. This score represents the predicted probability that the SQL is valid. During training, we employ the binary cross-entropy (BCE) loss to supervise the prediction, which is defined as follows:

$$\mathcal{L}_{\text{BCE}}(\hat{y}, y) = -[y \log(\hat{y}) + (1 - y) \log(1 - \hat{y})], \tag{7}$$

where $y \in \{0, 1\}$ denotes the ground-truth label indicating whether the SQL query is invalid ($y = 1$) or valid ($y = 0$). Notably, after model training is complete, the embeddings aggregated at each node of the logical plan serve as valuable sources of semantic information. Therefore, we can leverage the embeddings at logical plan nodes, together with the question embedding, to identify and localize erroneous sub-SQL segments produced by the LLM during the Text-to-SQL generation process. By providing targeted feedback based on the LP-level semantic signals, we can guide the LLM to recognize and correct specific errors in its intermediate outputs to iteratively refine its responses.

**Acceleration Strategies.** To mitigate the time overhead associated with context-guided property embedding, there are two acceleration strategies being employed: ❶ **KV Cache-based Schema Encoding.** Since the schema information remains unchanged for a given database throughout the training process, we leverage the key-value (KV) cache mechanism (Dao et al., 2022; Dao, 2024) to store intermediate representation results related to schema encoding. During repeated model invocations for the same database, previously computed KV cache entries of the schema can be directly reused, allowing the model to bypass redundant schema encoding computations. ❷ **Schema Information Compression.** In the context of SQL semantic validation, detailed attributes for each schema column—such as data type, nullability, and primary key status—exert minimal influence on the semantic interpretation required for most downstream tasks. Therefore, to reduce the token footprint associated with schema representation, we retain only essential information: table names and corresponding column names. By filtering out non-essential metadata, we not only decrease the input length but also accelerate downstream processing, without notable loss in semantic adequacy for the intended validation tasks.

## 5 EXPERIMENTS

In this section, we conduct comprehensive experiments on three Text-to-SQL datasets to evaluate the effectiveness of our method. Further details and experiment results are provided in Section E.

### 5.1 EXPERIMENTAL SETUP

**Datasets.** We employ three datasets to evaluate the performance of Text-to-SQL validation, including two cross-domain Text-to-SQL datasets, *BIRD* (Li et al., 2024c) and *Spider* (Yu et al., 2018), and one medical domain dataset, *EHRSQL* (Lee et al., 2022), which is derived from queries about Electronic Health Records (EHR). To investigate more complex schemas and SQL queries, we also use the *Spider 2.0* dataset (Lei et al., 2024), which is much more difficult than *BIRD* and *Spider* datasets. More details of the four datasets can be found in Section E.1.

**Methods and Baselines.** In our experiments, we utilize two embedding models, namely *Qwen3-Embedding-0.6B* (Zhang et al., 2025) and *embeddinggemma-300m* (Team et al., 2025), for embedding-based methods. For LLM-based methods, we use the corresponding LLMS *Qwen3-0.6B* (Yang et al., 2025) and *gemma-3-270m-it* (Team et al., 2025). To rigorously evaluate the effectiveness of our proposed approach, we compare HEROSQL against two categories of baselines: ❶ Baselines that treat SQL queries as plain text include Prompt, Chain-of-Thought (CoT) (Wei et al., 2022), ConfScore (Somov & Tutubalina, 2025), and COVE (Dhuliawala et al., 2024). ❷ To represent graph-based approaches which exploit the structural information of SQL queries, we select TED (Chen et al., 2023b), which leverages abstract syntax trees (ASTs) to model SQL query structure. Implementation details for all baseline methods are provided in Section D.1.

**Experimental Settings.** During training, we combine the training splits of the *BIRD* and *Spider* datasets to construct a unified base training dataset, and apply data augmentation techniques described in Section 4.2 to the dataset. Specifically, we first employ LLM-based data generation, followed by AST-driven sub-SQL augmentation, to expand the training data. For evaluation, we assess model performance on both in-domain datasets on *BIRD* and *Spider* and out-of-domain dataset *EHRSQL* and *Spider 2.0*. Notably, during testing, we use only the LLM-based data augmentation for each dataset to match the validation scenario in the real world, excluding samples derived from AST-based negative data augmentation to ensure consistency in evaluation. Detailed setting and evaluation of the experiments in Section E.2.

**Evaluation Metrics.** To measure semantic validation performance, we follow existing researches (Chen et al., 2023a) and adopt two threshold-independent metrics commonly used in binary classification: the *Area Under the Precision-Recall Curve* (AUPRC) and the *Area Under the Receiver Operating Characteristic Curve* (AUROC). These metrics provide a comprehensive assessment of model effectiveness regardless of the specific threshold chosen.

### 5.2 TEXT-TO-SQL VALIDATION RESULTS

As discussed, comprehensive experiments are conducted and the evaluation results on the three datasets are shown in Table 1. The results lead to the following observations:

**Outperforming SOTA Methods.** Our proposed HEROSQL outperforms all baseline models across three Text-to-SQL validation datasets, confirming its superior ability to detect semantic discrepancies between questions and generated SQL queries. When evaluated with the in-domain datasets *BIRD* and *Spider*, HEROSQL achieves the highest AUPRC and AUROC scores among all methods, with average improvements of **16.28%** (AUPRC) and **10.50%** (AUROC) over previous state-of-the-art models. One of the key reasons underlying this performance gain is that our method can effectively bridge global intent with local details. By capturing both the overarching semantic intent and the fine-grained local details within the SQL query, HEROSQL is able to perform more comprehensive and accurate cross-validation between the two modalities.

**Strong Semantic Validation Performance on Unseen Datasets.** As shown in Table 1, HEROSQL demonstrates strong semantic validation performance even on the clinical domain

Table 1: Performance comparison (%) on BIRD, Spider, EHRSQL and Spider 2.0 datasets. Results for methods with backbones of Qwen3-0.6B and Gemma-3-0.3B are presented in separate blocks for clarity. **Bold** indicates best results, and underline indicates second best.

| Method | In-Domain | | | | Out-of-Domain | | | |
| --- | --- | --- | --- | --- | --- | --- | --- | --- |
| | BIRD | | Spider | | EHRSQL | | Spider 2.0 | |
| | AUPRC | AUROC | AUPRC | AUROC | AUPRC | AUROC | AUPRC | AUROC |
| *Qwen3-0.6B* | | | | | | | | |
| Prompt | 60.85 | 57.25 | 40.86 | 58.94 | 84.72 | 58.86 | 88.78 | 51.80 |
| CoT | 61.01 | 55.31 | 39.58 | 50.98 | 84.13 | 59.61 | 89.15 | 48.32 |
| ConfScore | 56.59 | 51.43 | 34.13 | 50.65 | 80.66 | 50.39 | 90.15 | 44.77 |
| COVE | 57.67 | 54.40 | 38.09 | 54.78 | 85.53 | 56.54 | 90.75 | 61.86 |
| TED | 56.28 | 52.37 | 33.14 | 46.63 | 76.07 | 41.76 | 91.44 | 58.06 |
| HEROSQL | **67.39** | **61.51** | **51.92** | **67.32** | **89.07** | **69.53** | **92.59** | **64.32** |
| - w/o AST & LP | 62.66 | 59.23 | 49.28 | 61.15 | 88.23 | 63.24 | 90.83 | 59.25 |
| - w/o AST | 64.29 | 55.49 | 46.81 | 62.73 | 87.34 | 69.20 | 91.28 | 59.13 |
| - w/o NDA | 62.70 | 59.25 | 48.91 | 59.97 | 88.62 | 68.86 | 90.73 | 59.36 |
| *Gemma-3-0.3B* | | | | | | | | |
| Prompt | 54.13 | 48.47 | 34.26 | 49.75 | 78.86 | 44.79 | 88.30 | 44.58 |
| CoT | 54.72 | 49.84 | 34.28 | 49.96 | 81.73 | 54.11 | 89.22 | 50.00 |
| ConfScore | 54.79 | 50.53 | 35.77 | 50.18 | 78.48 | 46.14 | 90.79 | 52.34 |
| COVE | 55.02 | 50.46 | 34.25 | 49.78 | 81.58 | 51.79 | 88.51 | 45.86 |
| TED | 60.36 | 57.65 | 43.90 | 62.84 | 82.35 | 50.76 | 89.14 | 49.56 |
| HEROSQL | **67.10** | **63.48** | **51.11** | **69.26** | **85.48** | **65.92** | **91.55** | **59.85** |
| - w/o AST & LP | 66.71 | 61.38 | 50.39 | 67.82 | 85.10 | 62.08 | 90.13 | 54.49 |
| - w/o AST | 64.70 | 60.44 | 50.03 | 67.29 | 85.22 | 65.02 | 90.90 | 53.28 |
| - w/o NDA | 64.39 | 60.56 | 49.73 | 66.19 | 84.14 | 60.77 | 90.78 | 52.56 |

dataset *EHRSQL* and complex dataset *Spider 2.0*, whose training sets did not appear during the training process of HEROSQL. HEROSQL reaches an average of **3.97%** improvement in AUPRC and **19.34%** in AUROC on the out-of-domain dataset *EHRSQL*, highlighting its generalization ability without domain-specific fine-tuning. And on *Spider 2.0* dataset, HEROSQL achieves an improvement of **1.05%** in AUPRC and **9.16%** in AUROC, demonstrating that it maintains strong performance even in scenarios with more complex schemas and SQL queries. This further confirms the robustness and generalization capability of HEROSQL across different domains and task complexities , making it a promising solution for practical deployment in diverse Text-to-SQL applications.

## 5.3 TEXT-TO-SQL ERROR CORRECTION AIDED BY HEROSQL

We conduct Text-to-SQL error correction experiments on the DEV set of the *BIRD* dataset to evaluate the effectiveness of HEROSQL on improving the end-to-end performance of a Text-to-SQL pipeline. Our experiment compares end-to-end Text-to-SQL without self-correction, Prompt-based error detection method, and HEROSQL-based error detection method with and without LP-level feedback signals for error correction. The experiments are evaluated using two backbone models, *gpt-4o* and *qwen-plus*, and the accuracy across three difficulty levels (simple, moderate, challenging) and the total accuracy are reported.

As shown in Table 2, applying HEROSQL-based error detection improves the end-to-end performance for Text-to-SQL tasks. For both non-thinking LLM backbone *gpt-4o* and reasoning LLM backbone *qwen-plus*, HEROSQL with LP-level feedback achieves the highest total accuracy, outperforming other baseline methods. Specially, HEROSQL-based verifier can significantly improve the performance of LLM for challenge tasks on BIRD dataset, gaining an improvement of **8.97%**

Table 2: Evaluation results with both *gpt-4o* and *qwen-plus* backbones for different Text-to-SQL error correction methods on the DEV set of *BIRD* dataset under each difficult levels. **Bold** indicates best results, and underline indicates runner-up results.

| Method | Simple | Moderate | Challenging | Total |
|---|---|---|---|---|
| *gpt-4o* | | | | |
| W/o self-correction | **60.22** | 40.09 | 31.72 | 51.43 |
| Prompt-base verifier | 57.51 | 38.36 | 29.66 | 49.09 |
| HEROSQL-base verifier w/o LP-level feedback | 59.24 | 42.03 | **40.69** | 52.28 |
| HEROSQL-base verifier with LP-level feedback | **60.22** | **43.97** | 37.93 | **53.19** |
| *qwen-plus* | | | | |
| W/o self-correction | 66.16 | 50.65 | 49.66 | 59.91 |
| Prompt-base verifier | 64.00 | 48.28 | 49.66 | 57.89 |
| HEROSQL-base verifier w/o LP-level feedback | 65.84 | 53.02 | **51.72** | 60.63 |
| HEROSQL-base verifier with LP-level feedback | **66.81** | **54.31** | 51.34 | **61.47** |

with no self-correction baselines. This improvement is mainly due to the fact that HEROSQL can effectively detect potential errors in the generated SQL, prompting the non-thinking model to reflect on the previously generated SQL and thereby correct the earlier errors. Notably, introducing LP-level feedback further boosts performance. These results demonstrate that accurate error detection and LP-level guidance feedback are effective in enhancing the end-to-end performance of Text-to-SQL pipeline. We also provide some case studies about Text-to-SQL error correction aided by HEROSQL in Appendix C.

## 5.4 ABLATION STUDY

To validate the effectiveness of each component in HEROSQL, we conduct three ablation studies within our method. ❶ HEROSQL *w/o NDA* model is trained on the dataset without AST-based negative data augmentation, while ❷ HEROSQL *w/o AST* is the model that encodes the logical plan expression directly as text, without expanding it, and ❸ HEROSQL *w/o AST & LP* is the model just takes the entire SQL query as input.

The results of our ablation study, as shown in Table 1, demonstrate the effectiveness of each component in HEROSQL. Removing NDA leads to a consistent decrease across all evaluation metrics, indicating that AST-based negative data augmentation helps the model better distinguish subtle semantic errors. When the AST is excluded, the model mainly captures global intent from LP but loses the ability to understand important structural and syntactic details. When both AST and LP are removed and the model is trained solely on plain text SQL queries, it becomes difficult for the model to capture either global intent or local grammatical and semantic details, resulting in performance decay. These results verify the effectiveness of our approach in addressing challenges *#C1* and *#C2*.

## 6 CONCLUSIONS AND FUTURE WORKS

In this paper, we propose HEROSQL: a hierarchical representation framework that uses Logical Plans to capture high-level computational logic and data flow (global intent) & Abstract Syntax Trees to model fine-grained schema details, and employs a Nested Message Passing Neural Network to aggregate schema-guided embeddings across relational AST/LP nodes. Moreover, an Adaptive sub-SQL Augmentation Strategy is introduced to generate large-scale syntactically valid but semantically incorrect negative samples via AST perturbations, thereby mitigating fine-grained data scarcity. Experiments on both in-domain datasets and out-of-domain datasets show that HEROSQL outperforms baselines in identifying semantic inconsistencies, which improves the reliability and interpretability of Text-to-SQL systems and reduces human verification costs. For future work, we will enhance the granularity of SQL validation by adding a fine-grained sub-SQL classification head that can not only detect semantic incorrectness in sub-SQL fragments but also classify specific error types (e.g., filter condition mismatches, aggregate function misselection, or join relation errors), achieving more precise and targeted semantic validation for Text-to-SQL queries.

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

APPENDIX TABLE OF CONTENTS

## A    NOTATIONS TABLE

The main notations in this paper are summarized in Table 3.

Table 3: Notations in HEROSQL

| Notation | Definition |
|---|---|
| $q$ | natural language question |
| $s$ | SQL query corresponding to question $q$ |
| $\hat{y}$ | predicted validation probability score |
| $y$ | validation label corresponding to question-SQL pair |
| $f$ | validation function |
| $\mathcal{LP}$ | logical plan |
| $\mathcal{G}_{\mathcal{LP}}$ | directed acyclic graph of logical plan $\mathcal{LP}$ |
| $V_{\mathcal{LP}}$ | node sets of logical plan $\mathcal{G}_{\mathcal{LP}}$ |
| $E_{\mathcal{LP}}$ | edge sets of logical plan $\mathcal{G}_{\mathcal{LP}}$ |
| $v_i^{\mathcal{LP}}$ | node of logical plan $\mathcal{G}_{\mathcal{LP}}$ |
| $o_i$ | relational operator of node $v_i^{\mathcal{L}}$ |
| $a_i$ | corresponding attributes of node $v_i^{\mathcal{L}}$ |
| $\mathcal{A}_i$ | Abstract Syntax Tree for node $i$ of $\mathcal{LP}$ |
| $\mathcal{G}_{\mathcal{AST}}$ | directed acyclic graph of Abstract Synatx Tree $\mathcal{LP}$ |
| $V_{\mathcal{AST}}$ | node sets of AST $\mathcal{G}_{\mathcal{AST}}$ |
| $E_{\mathcal{AST}}$ | edge sets of AST $\mathcal{G}_{\mathcal{AST}}$ |
| $v_j^{\mathcal{A}}$ | node j of AST $\mathcal{G}_{\mathcal{AST}}$ |
| $c_j^{\mathcal{A}}$ | atom syntax of $v_j^{\mathcal{A}_i}$ |
| $\mathcal{R}_{SQL}$ | hierarchical intermediate representation |
| $g$ | LLM-based embedding model |
| $H$ | sequence of hidden states output by $g$ |
| $h_v^{(t)}$ | representation of AST node $v \in V_{\mathcal{A}_i}$ at message passing step $t$ |
| $\mathcal{N}(v)$ | neighbor set of node $v$ in AST $G_i$ |
| $T_{\text{ast}}$ | number of message passing steps in AST-level MPNN |
| $h_i$ | aggregated embedding of LP node $i$ |
| $h_i^{(t)}$ | representation of LP node $i$ at message passing step $t$ in logical-plan-level |
| $\mathcal{N}_L(i)$ | neighbor set of LP node $i$ in the logical plan graph |
| $T_{\text{LP}}$ | number of message passing steps in logical-plan-level |
| $\mathcal{D}_{\text{gold}}$ | ground truth Text-to-SQL dataset |
| $\mathcal{D}_{\text{LLM}}$ | synthetic dataset with LLM-based data augmentation |
| $\mathcal{D}_{\text{AST}}$ | synthetic dataset with AST-based negtive data augmentation |
| $\mathbb{N}^+$ | index set of natural numbers (used to denote the size of $\mathcal{D}_{\text{gold}}$) |
| $\mathcal{T}$ | set of AST-level transformation rules |
| $h_{\text{SQL}}$ | SQL query embedding |
| $h_{\text{question}}$ | question embedding |
| $h_{\text{hadamard}}$ | Hadamard fusion embedding |
| $\mathbf{h}$ | fused embedding vector |

## B    THE CHOICE OF EMBEDDING MODEL ARCHITECTURE

In our work, we use embedding models with **decoder-only architecture** as the embedding backbone, rather than the encoder-only architecture that is more common in text encoding tasks. The choice of using decoder-only embedding models built upon large language models is because these models offer several distinct advantages.

❶ **Improved generalization ability**: With the development of large language models that are pretrained on vast and diverse corpora with next-token prediction objectives, these **decoder-only models can produce richer contextual representations**, enabling better transferability to downstream tasks and more robust performance across heterogeneous datasets. Embedding models like *Qwen3-*

*Embedding* (Zhang et al., 2025) built upon *Qwen3* series (Yang et al., 2025) and *EmbeddingGemma* built upon *Gemma 3* series (Team et al., 2025) can have a better understanding of the unseen context in the training dataset. In contrast, **open-source encoder-only models typically have far fewer parameters** than commonly used decoder-only models, which means they usually contain less knowledge. For example, even the frequently used encoder-only architecture model *GTE-large* (Li et al., 2023b) and *BGE-large-en* (Xiao et al., 2024) have only around 0.3B parameters. As a result, these encoder-only models have relatively limited capacity to encode rich task-related knowledge. However, decoder-only models are available at significantly larger scales. Although we mainly use smaller models in our current experiments for computational efficiency, they can be readily replaced with larger variants when tackling more complex tasks.

❷ **Enhanced context length**: Modern decoder-only large language models are optimized to process significantly longer input sequences than their encoder-only counterparts. For example, *Qwen3-Embedding* series models support a maximum input sequence length of 32k tokens (Zhang et al., 2025), while *BERT* series models are limited to a context length of only 512 tokens (Devlin et al., 2019). The longer context allows models with decoder-only architecture to achieve richer semantic understanding over extended passages of text, which is crucial in realistic scenarios where a complex user query, database schema, and additional domain knowledge must be included in the context. Longer context windows make it easier to incorporate more detailed schema information, examples, and auxiliary hints, which is especially important in our HEROSQL framework.

To further verify these these, we conduct a experiment on *BIRD*, *Spider* and *EHRSQL* datasets with both encoder-only architecture models and decoder-only architecture models. In this experiment, we compare the performance of HEROSQL under difference embedding models, including ones with comparable parameter sizes (*Qwen3-0.6B* and *Gemma-3-0.3B*) as well as a larger variant (*Qwen3-4B*), against strong encoder-only baselines (*GTE-large* and *BGE-large-en*) within the HEROSQL framework. The results are shown in Table 4

Table 4: Comparison of the performance of HEROSQL under different embedding models.

| Embedding Model | BIRD | | Spider | | EHRSQL | |
|---|---|---|---|---|---|---|
| | AUPRC | AUROC | AUPRC | AUROC | AUPRC | AUROC |
| GTE-large | 58.74 | 54.39 | 43.12 | 61.28 | 79.86 | 46.51 |
| BGE-large-en | 62.01 | 57.78 | 42.87 | 61.87 | 81.41 | 49.34 |
| Gemma-3-0.3b | 67.10 | 63.48 | 51.11 | 69.26 | 85.48 | 65.92 |
| Qwen3-0.6b | 67.39 | 61.51 | 51.92 | 67.32 | 89.07 | 69.53 |
| Qwen3-4b | **71.18** | **64.17** | **53.52** | **69.78** | **89.28** | **72.03** |

Results from Table 4 show that decoder-only models with a parameter scale comparable to encoder-only models already achieve consistently better performance across all three datasets in HEROSQL. Moreover, scaling up the decoder-only model (e.g., from 0.6B to 4B) further improves performance, indicating a favorable scaling behavior for decoder-only embeddings in our setting.

## C  CASE STUDY ON TEXT-TO-SQL ERROR CORRECTION AIDED BY HEROSQL

In this section, we present two detailed case studies selected from BIRD dataset to illustrate the problem-solving capability of HEROSQL when correcting erroneous SQL queries generated by a large language model (LLM). These examples demonstrate how HEROSQL effectively diagnoses the precise location and nature of the error in both the raw SQL and its logical plan representation, ultimately leading to the generation of the correct SQL.

As shown in Table 5, the initial LLM output (HEROSQL Output) mistakenly adds WHERE T1. rtype = 'Reading'. HEROSQL identifies this as an incorrect filter placement via the Logical Plan, where LogicalFilter(condition=[=(rtype, 'Reading')]) is applied after the JOIN. The analysis highlights that the satscores table inherently contains only reading scores (as indicated by the schema not being fully utilized by the initial query logic or a subtle data characteristic), making the ex-

plicit WHERE clause redundant and potentially inefficient if multiple rtype values were involved. The feedback correctly advises its removal, demonstrating that HEROSQL can detect and correct unnecessary/redundant operations, guiding the LLM to the concise and correct Ground Truth query.

Table 5: Case study 1 on Text-to-SQL error correction process with the help of HEROSQL.

| Question | For the school with the highest average score in Reading in the SAT test, what is its FRPM count for students aged 5-17 ? |
|---|---|
| **Ground Truth** | SELECT T2.'FRPM Count (Ages 5-17)' 
 FROM satscores AS T1 
 INNER JOIN frpm AS T2 
 ON T1.cds = T2.CDSCode 
 ORDER BY T1.AvgScrRead DESC 
 LIMIT 1; |
| **HEROSQL Output** | **\<LLM generate a error SQL\>** 
 SELECT T2.'FRPM Count (Ages 5-17)' 
 FROM satscores AS T1 
 INNER JOIN frpm AS T2 
 ON T1.cds = T2.CDSCode 
 **WHERE T1.rtype = 'Reading'** 
 ORDER BY T1.AvgScrRead DESC 
 LIMIT 1; 

 \<Anomalous node in Logical Plan\> 
 LogicalProject(FRPM Count (Ages 5-17)=['FRPM Count (Ages 5-17)']) 
 LogicalSort(sort0=[AvgScrRead], dir0=[DESC], fetch=[1]) 
 LogicalProject(FRPM Count (Ages 5-17)=['FRPM Count (Ages 5-17)'], AvgScrRead=[AvgScrRead]) 
 **LogicalFilter(condition=[=(rtype, 'Reading')])** 
 LogicalJoin(condition=[=(cds, CDSCode)], joinType=[inner]) 
 JdbcTableScan(table=[[main, satscores]]) 
 JdbcTableScan(table=[[main, frpm]]); 

 \<LLM analyze based on error part\> 
 The mistake occurs because the filter 'WHERE rtype = 'Reading'' is applied after the join in the predicted query and logical plan, rather than restricting the SAT scores table before aggregation or selection. In the logical plan, the 'LogicalFilter(condition=[=(rtype, 'Reading')])' is placed after the join, which is unnecessary and inefficient because it forces the database to combine all rows from 'satscores' and 'frpm' before filtering only the Reading scores. This can lead to unnecessary data processing, larger intermediate result sets, and potentially incorrect selection if multiple 'rtype' values exist for the same school. The core issue is that the filter should target the SAT scores table **before the join**, ensuring that only Reading scores are considered when identifying the school with the highest average, making the query both correct and efficient. 

 \<LLM generate a correct SQL\> 
 SELECT T2.'FRPM Count (Ages 5-17)' 
 FROM satscores AS T1 
 INNER JOIN frpm AS T2 
 ON T1.cds = T2.CDSCode 
 ORDER BY T1.AvgScrRead DESC 
 LIMIT 1; |

As shown in Table 6, the LLM-generated SQL incorrectly uses WHERE schools.City = 'Fres', which the Logical Plan representation clearly shows as LogicalFilter(condition=[=(City, 'Fres')]). HEROSQL's analysis pinpoints a critical semantic error: the query filters by a non-existent City name ('Fres') instead of the required County Name ('Fresno'). This deep understanding of the question's intent versus the query's implementation allows HEROSQL to generate feedback that corrects both the column name (City to County Name) and the predicate value ('Fres' to 'Fresno').

This detailed, contextual error analysis enables the LLM to generate the final, logically correct SQL query that accurately addresses the user's intent.

As shown in Table 7, the LLM-generated SQL incorrectly uses an INNER JOIN between schools and satscores, as highlighted in the Logical Plan with LogicalJoin(condition=[=(CDSCode, cds)], joinType=[inner]). HEROSQL 's analysis identifies a critical semantic mistake: the query assumes satscores.AvgScrWrite already contains the average writing score, ignoring the question's requirement to compute the average over schools filtered by opening and closing dates. Moreover, the inner join would exclude schools without scores, leading to incomplete results. By understanding the intended aggregation and the need to preserve all eligible schools, HEROSQL generates feedback that corrects the join type to LEFT JOIN and clarifies that an explicit AVG aggregation is required. This detailed reasoning allows the LLM to produce a corrected SQL query that accurately computes the average writing scores for the specified subset of schools, while retaining optional phone numbers.

# D  IMPLEMENTATION DETAILS

## D.1  BASELINE IMPLEMENTATION

- **Prompt.** We directly prompt a large language model (LLM) to determine whether a natural language question and an SQL query are semantically aligned. Given a user's natural language question and the corresponding database schema, we request the LLM to check whether the generated SQL query is semantically correct based on a list of validation rules. We provide the validation prompt in Section D.5
- **CoT.** We extend Prompt method by applying chain-of-thought prompting (Wei et al., 2022), allowing the model to reason step by step before deciding whether the text and SQL match. Specifically, instead of requiring the LLM to provide a direct judgment, we instruct the model to explicitly enumerate the semantic intent expressed in the user's question, to analyze the components of the SQL query, and to compare their correspondence according to a set of validation criteria.
- **ConfScore.** We build on the confidence-based evaluation method of Somov & Tutubalina (2025), originally proposed for SQL generation. To adapt it to the SQL verification setting, we further draw on the prompting strategy of Kim et al. (2023), enabling the model to take both the text and candidate SQL as input and produce an explanation whose confidence is then used to assess correctness.
- **COVE.** Inspired by Dhuliawala et al. (2024), COVE reduces hallucinations through verification and correction. While the original pipeline involves generating an initial answer and then validating it, in our SQL verification setting we skip the answer generation step and directly apply the subsequent stages: (i) planning verification questions from the text–SQL pair, (ii) answering these questions independently, and (iii) aggregating the verification results to assess whether the SQL is correct.
- **TED.** Text-to-SQL Error Detector (TED) (Chen et al., 2023b) is a parser-agnostic model that jointly encodes natural language questions and SQL queries using CodeBERT (Feng et al., 2020), while incorporating graph neural networks (GNN) to capture structural information. By modeling dependency trees of questions and abstract syntax trees (ASTs) of SQL queries, TED explicitly learns the semantic alignment between text and query structures, enabling more accurate detection of mismatches.

## D.2  DETAILS ABOUT AST-LEVEL TRANSFORMATION RULES

To construct a series of challenging negative samples from valid SQL queries, we define a set of AST-level transformation rules $\mathcal{T}$ to generate samples that are grammatically correct but semantically incorrect, which is introduced in Section 4.2.2. All transformation rules leveraged in our work are shown below.

- **Rule 1: Operator Inversion.** Many SQL queries rely on logical and comparison operators to specify selection conditions. To perturb the query's semantics while retaining grammatical validity, we invert operators at the AST level. For example, greater-than operators ($>$) are replaced with less-than-or-equal-to ($\leq$), and logical connectors such as AND are substituted with OR. This transformation ensures the resulting query is syntactically correct, but alters the retrieval logic to produce incorrect results.

Table 6: Case study 2 on Text-to-SQL error correction process with the help of HEROSQL.

| Question | How many schools in Fresno (directly funded) have number of test takers not more than 250? |
|---|---|
| **Ground Truth** | SELECT COUNT(T1.CDSCode)
FROM frpm AS T1
INNER JOIN satscores AS T2
ON T1.CDSCode = T2.cds
WHERE T1.'Charter Funding Type' = 'Directly funded'
AND T1.'County Name' = 'Fresno'
AND T2.NumTstTakr <= 250; |
| **HEROSQL Output** | \<LLM generate a error SQL>
SELECT COUNT(*)
FROM schools
JOIN satscores
ON schools.CDSCode = satscores.cds
**WHERE schools.City = 'Fres'**
AND schools.FundingType = 'Directly Funded'
AND satscores.NumTstTakr <= 250; |
|  | \<Anomalous node in Logical Plan>
LogicalAggregate(group=[], EXPR$0=[COUNT()])
**LogicalFilter(condition=[AND(=(City, 'Fres'),**
=(FundingType, 'Directly Funded'),
<=(NumTstTakr, 250))])
LogicalJoin(condition=[=(CDSCode, cds)], joinType=[inner])
JdbcTableScan(table=[[main, schools]])
JdbcTableScan(table=[[main, satscores]]); |
|  | \<LLM analyze based on error part>
The mistake occurs because the query filters 'City = 'Fres'', whereas the user's intent is to count schools in Fresno County, not a city named "Fres." In the logical plan, this corresponds to the 'LogicalFilter(condition=[=(City, 'Fres')]...)' applied before the join. This condition is incorrect because 'City' is the wrong column; the correct filter should be ''County Name' = 'Fresno''. As a result, the query may return zero rows or only schools in a non-existent city "Fres," completely missing schools that are actually in Fresno County. The redundant or incorrect part is the 'City = 'Fres'' predicate, which should be replaced with ''County Name' = 'Fresno'' to correctly reflect the intended data selection. Without fixing this, the COUNT() will not represent the number of directly funded schools in Fresno County with ≤ 250 test takers. |
|  | \<LLM generate a correct SQL>
SELECT COUNT(*)
FROM schools
JOIN satscores
ON schools.CDSCode = satscores.cds
WHERE schools.'County Name' = 'Fresno'
AND schools.FundingType = 'Directly Funded'
AND satscores.NumTstTakr <= 250; |

Table 7: Case study 3 on Text-to-SQL error correction process with the help of HEROSQL.

| Question | What is the average score in writing for the schools that were opened after 1991 or closed before 2000? List the school names along with the score. Also, list the communication number of the schools if there is any. Communication number refers to phone number. |
|---|---|
| **Ground Truth** | SELECT s.School, s.Phone, satscores.AvgScrWrite
FROM schools AS s
LEFT JOIN satscores
ON s.CDSCode = satscores.cds
WHERE s.OpenDate > '1991-12-31'
OR s.ClosedDate < '2000-01-01' |
| **HEROSQL Output** | \<LLM generate a error SQL\>
SELECT s.School, s.Phone, satscores.AvgScrWrite
FROM schools s
**JOIN satscores**
ON s.CDSCode = satscores.cds
WHERE s.OpenDate > '1991-12-31'
OR s.ClosedDate < '2000-01-01' |
| | \<Anomalous node in Logical Plan\>
LogicalProject(SCHOOL=[School], PHONE=[Phone], AVGSCR-WRITE=[AvgScrWrite])
LogicalFilter(condition=[OR(>(OpenDate, '1991-12-31'), <(ClosedDate, '2000-01-01'))])
**LogicalJoin(condition=[=(CDSCode, cds)], joinType=[inner])**
JdbcTableScan(table=[[main, schools]])
JdbcTableScan(table=[[main, satscores]]) |
| | \<LLM analyze based on error part\>
The mistake occurs because the predicted SQL and logical plan treat 'satscores.AvgScrWrite' as if it already contains the "average writing score," whereas the question actually asks for calculating the average writing score for the subset of schools that meet the date criteria. In the query, the 'INNER JOIN' between 'schools' and 'satscores' is correct for matching CDS codes, but it does not account for schools that might lack scores—here a 'LEFT JOIN' would ensure all eligible schools are included even if no scores exist. Moreover, the query lacks an 'AVG(satscores.WriteScore)' aggregation over the filtered subset; directly selecting 'AvgScrWrite' is incorrect and redundant. The implication is that the query may either exclude schools without scores (due to the inner join) or return precomputed averages that do not reflect the filtered date condition, producing inaccurate results. In short, the query should use a 'LEFT JOIN' and an explicit 'AVG' aggregation over the writing scores of schools that satisfy the opening/closing date condition. |
| | \<LLM generate a correct SQL\>
SELECT s.School, s.Phone, satscores.AvgScrWrite
FROM schools s
LEFT JOIN satscores
ON s.CDSCode = satscores.cds
WHERE s.OpenDate > '1991-12-31'
OR s.ClosedDate < '2000-01-01' |

- **Rule 2: Identifier Substitution.** In this rule, we randomly select field identifiers from the same database schema and replace the original field names in the SQL query's AST. For instance, replacing column name 'salary' with 'age', or replacing table name 'department' with 'location'. The replacement fields are drawn from the candidate column names pool or table names pool to ensure grammatical correctness, but disrupt the intended query semantics.

- **Rule 3: Constant Replacement.** We modify constant values appearing in the SQL conditions by substituting them with other constants from the database. Suppose a condition checks for "age = 30"; we may replace 30 with a different valid value such as 40. The new constants are selected to be plausible for the context but alter the outcome of the query.

- **Rule 4: Aggregation Function Mutation.** Aggregation functions define how rows are combined in SQL queries. We alter these functions by replacing one aggregation operator with another, e.g., substituting AVG with MAX, MIN, or COUNT. This transformation keeps the query structurally intact but changes its fundamental semantic intention.

These rules constitute a comprehensive transformation set designed for AST-based SQL augmentation. By applying $\mathcal{T}$ to source queries, we generate a variety of negative samples that challenge models on fine-grained semantic understanding without violating SQL syntax. The full instantiation of each rule is implemented over the SQL AST, allowing flexible and systematic query perturbation for downstream training and evaluation.

### D.3 CONSTRUCTION AND SPLITTING STRATEGY OF DATASETS

As for the datasets, we follow the original train/test splits for all three datasets: *BIRD*, *Spider*, and *EHRSQL*. During training, we combine the training splits of the *BIRD* and *Spider* datasets to construct a unified base training dataset. Building upon this merged dataset, we apply the data augmentation techniques described in Section 4.2.1. Specifically, we first employ LLM-based data validation generation, followed by AST-driven sub-SQL augmentation strategy that introduced in Section 4.2.2, to expand the training data. To balance the ratio of positive and negative samples in the training data more effectively, we imposed a restriction on the number of augmented samples during the AST-driven sub-SQL augmentation process, ensuring that the ratio of positive to negative samples in the final training set could reach 1:1. The final augmented training set thus integrates both LLM-generated validation data and augmentated semantic error samples, and we split the final data into training and validation sets using a ratio of 80% to 20% .

Notably, **all samples generated by the AST-driven sub-SQL augmentation strategy are truely semantically incorrect**. Concretely, our training set is constructed from SQL queries that have known ground-truth SQL counterparts, which can be executed to obtain ground-truth execution results. After applying AST perturbations, we execute each perturbed SQL query and compare its execution result with that of the corresponding ground-truth query. Only perturbed SQL queries whose execution results differ from the original are retained as semantically incorrect samples in the augmented dataset. This strict, result-based filtering, ensemble reverse reject-sampling, substantially reduces the likelihood of including false negatives and ensures that retained perturbations correspond to genuine semantic or logical errors.

Regarding overfitting to synthetic mistakes, the AST-perturbed negative samples are always used in conjunction with the original (question, SQL) pairs and other negative examples in our augmented corpus. In practice, this means the model is exposed to a diverse mixture of real and systematically perturbed errors, rather than being trained solely on a narrow class of synthetic perturbations. This diversity, together with the execution-based filtering, helps mitigate overfitting and encourages the model to learn robust semantic distinctions rather than artifacts of a particular perturbation scheme.

In the evaluation process, we assess model performance on both in-domain and out-of-domain datasets. The in-domain performance is measured on the *BIRD* and *Spider* test sets, while out-of-domain generalization is evaluated using the *EHRSQL* dataset. Notably, during testing, we use only the LLM-based validation data augmentation for each dataset, excluding samples derived from AST-driven sub-SQL augmentation to ensure consistency in evaluation.

## D.4 THRESHOLD SELECTION FOR REAL-WORLD APPLICATIONS

The binary validator decision threshold in our Text-to-SQL pipeline should be chosen according to the requirements of the downstream applications. Improved AUROC/AUPRC implies a better Pareto frontier, enabling practitioners to select more favorable precision–recall trade-offs for different risk profiles. Concretely:

- In scenarios where *false positives (FP)* are more harmful (e.g. over-correcting originally correct SQL and potentially regressing performance), one should choose a higher threshold that prioritizes precision, even at the cost of recall. In practice, this can be done by selecting the largest threshold at which precision stays above a desired target (e.g., $\geq 95\%$).

- In scenarios where *false negatives (FN)* are more costly (e.g. it is critical not to miss any erroneous SQL being executed), one may prefer a lower threshold that emphasizes high recall, while allowing precision to vary within an acceptable range.

In our experiments, we report results using the threshold that maximizes the F1 score on a validation set, providing a balanced operating point between precision and recall. This is a standard choice when there is no application-specific preference between FP and FN. We also illustrate that, under a reasonable threshold for error detection and targeted correction, integrating our validator can improve end-to-end Text-to-SQL accuracy.

## D.5 PROMPT USED IN THE FRAMEWORK

In this section, we provide a detailed introduction to the prompts used in our framework.

---

**LLM-based Data Augmentation Prompt**

You are a helpful data analysis expert. Given a user question and the corresponding database schema, please output a syntactically correct and efficient SQL query that fully answers the user's request.

The database schema is as follows:
{schema}

Write SQL for the following question:
{question}

Please output the SQL query directly. Do NOT output any other comments.

---

---

### LLM Verification Prompt

**[Role]**
You are an expert specializing in SQL query verification, dedicated to the task of determining whether a given SQL can solve the user's problem, and providing high-quality analysis results based on specific rules. You possess strong knowledge of SQL syntax, understanding of database structures, and natural language processing abilities, allowing you to accurately judge the match between SQL queries and user questions.

**[Workflow]**

1. **Input Acquisition**
   **Receive** three essential pieces of information from the user:
   - **User Question (question)**
   - **Database Structure (schema)**
   - **SQL Query Statement**

   Ensure successful parsing and prepare for detailed analysis.

2. **Analysis / Processing Logic**
   2.1. **Question Analysis:**
   Analyze the semantic intent of the user question, identifying key **entities**, **attributes**, and **query conditions**.
   2.2. **Database Structure Analysis:**
   Understand the **table structure**, **field types**, and **relationships** within the schema. Determine which tables and fields are relevant to the user question.
   2.3. **SQL Syntax Check:**
   Verify the syntactic correctness of the SQL query, including:
   - keyword usage
   - table and field references
   - JOIN syntax, etc.
   2.4. **Semantic Matching Analysis:**
   - **SELECT** clause: Check if all fields required to answer the question are included.
   - **FROM** & **JOIN** clauses: Ensure all relevant tables are referenced.
   - **WHERE** clause: Confirm inclusion of all necessary filtering conditions.
   - **GROUP BY**, **HAVING**, **ORDER BY**, etc.: Verify compliance with the question's requirements.
   2.5. **Result Expectation Analysis:**
   Infer whether the result set after SQL execution can directly answer the user question, or if further processing is needed.

3. **Result Output**
   Based on the analysis results, provide a clear judgment:

   

   | approve | or | reject |
   
   

   *Answer "approve" or "reject" only, without any additional explanation.*

**[Question]**
{question}

**[Database Schema]**
{schema}

**[SQL]**
{SQL}

---

---

**LLM Analyze Based on Error Part Prompt**

You are an expert SQL analyst. Carefully examine the given SQL query, its logical plan, and the described mistake. Provide a detailed, step-by-step explanation in English.

Question:
{question}

Predicted SQL query:
{pred_sql}

Predicted logical plan:
{pred_plan}

Identified mistake of logical plan:
{pred_mistake}

Instructions for analysis:
1. Explain clearly why the mistake occurs.
2. Identify which part of the query is incorrect, redundant, or unnecessary.
3. Describe the implications of this mistake on query execution or results.
4. Provide a concise, understandable explanation suitable for someone familiar with SQL and query planning.

Output:
A detailed error analysis, focusing only on the predicted SQL and its logical plan. Please generate a short paragraph.

---

# E EXPERIMENTS DETAILS

## E.1 STATISTICAL INFORMATION OF DATASETS

Table 8: Overview of datasets used in our experiments.

| Dataset | Domain | #DB | #Table/DB | Train | Dev | Test |
|---|---|---|---|---|---|---|
| BIRD (Li et al., 2024c) | 37 | 95 | 7.3 | 9,428 | 1,534 | 1,789 |
| Spider (Yu et al., 2018) | 138 | 200 | 5.1 | 7,862 | 1,831 | 2,147 |
| EHRSQL (Lee et al., 2024) | 1 | 1 | 17 | 5,124 | 1,163 | 1,167 |
| Spider 2.0 (Saparina & Lapata, 2024) | / | 213 | 7.5 | / | / | 632 |
| Ambrosia (Saparina & Lapata, 2024) | 16 | 846 | 5.0 | / | / | 1,277 |

In our work, five different datasets are taken to evaluate Text-to-SQL validation performance: two general-purpose datasets, *BIRD* (Li et al., 2024c) and *Spider* (Yu et al., 2018), one practical medical dataset *EHRSQL* (Lee et al., 2022; 2024), one challenge dataset *Spider 2.0* (Lei et al., 2024) and one ambiguous dataset *Ambrosia* (Saparina & Lapata, 2024).

- **BIRD** (Li et al., 2024c) is a large-scale cross-domain benchmark designed for real-world Text-to-SQL tasks. It contains 12751 questions and their corresponding SQL queries, grounded in 95 databases across 37 domains. The dataset is divided into a training set with 9,428 instances, a development set with 1,534 instances, and a test set with 1,789 instances. BIRD's unique focus on large-scale databases and external knowledge makes it a challenging benchmark for real-world Text-to-SQL tasks.

- **Spider** (Yu et al., 2018) is also a cross-domain Text-to-SQL dataset. It contains 10,181 questions and 5,693 unique complex SQL queries across 200 databases across 138 domains.The dataset is evaluated under two settings: (i) example split, where 7,862/1,831/2,147 questions are randomly assigned to train/dev/test, allowing questions from the same database to appear across splits; and

(ii) database split, where 206 databases are divided into 130/36/40 for train/dev/test, ensuring all questions from a database remain within the same split. Spider poses greater challenges than prior text-to-SQL datasets due to its large number of complex queries (e.g., joins, nesting, grouping, ordering) and its requirement for cross-domain generalization to unseen database schema.

- **EHRSQL** (Lee et al., 2022; 2024) is a Text-to-SQL dataset focused on electronic health records (EHR), grounded in real-world database, MIMIC-IV (Johnson et al., 2023). Specifically, we use the EHRSQL 2024 dataset in our experiments. The dataset is split into a training set with 5124 questions, a development set with 1163 questions, and a test set with 1167 questions. A unique feature is the inclusion of unanswerable questions, which requires models to leverage external knowledge to determine question validity.

- **Spider 2.0** (Lei et al., 2024) is a large-scale, cross-domain Text-to-SQL benchmark that significantly scales up in database size, query complexity, and SQL dialect diversity compared to its predecessor. It comprises 632 test examples across 213 databases, with an average of 743.5 columns per database and 148.3 tokens per SQL query, reflecting a high degree of schema complexity and query length. Each SQL query involves an average of 7.1 functions, indicating extensive use of advanced SQL constructs. The databases span a wide range of real-world domains and include diverse data types like JSON, STRUCT, GEOGRAPHY, and TIMESTAMP across multiple SQL dialects, which include BigQuery, Snowflake and SQLite. Notably, Spider 2.0 databases are TB-scale, with an average of 5.27 billion rows, presenting unprecedented challenges in scale, schema understanding, and cross-domain generalization for Text-to-SQL systems. In our experiments, we evaluate models under the Spider 2.0-Lite subset using SQLite as the database engine.

- **Ambrosia** (Saparina & Lapata, 2024) is a cross-domain benchmark dedicated to parsing **ambiguous** natural-language questions into SQL. It contains 1,277 ambiguous questions, 2,965 corresponding SQL interpretations, and 846 multi-table databases spanning 16 realistic domains (e.g., Banking, Entertainment, Healthcare). Each ambiguous question is paired with all valid, human-verified SQL queries that arise from scope, attachment, or vagueness ambiguities, yielding 2–3 gold queries per question. The dataset is split 90 % for zero shot evaluation and 10 % for few-shot demonstration; questions are grouped by ambiguity type rather than by database to preserve linguistic variety. Compared with prior text-to-SQL corpora, Ambrosia introduces the first large-scale testbed where ambiguity persists even when the full schema and content are known, requiring models to explicitly recognize and generate multiple correct interpretations instead of a single canonical query.

In our experiments, the class distribution of training set is balanced with 1:1 ration because we applied negative-sample augmentation strategy for the training set. For the evaluation set, we analyzed the class distribution in each dataset used in our main experiments.

Table 9: Class distribution of test sets across datasets

| Dataset | #Correct | #Incorrect | Positive Ratio (%) |
|---------|----------|------------|--------------------|
| BIRD | 812 | 983 | 54.76 |
| Spider | 1019 | 532 | 34.30 |
| EHRSQL | 141 | 579 | 80.41 |

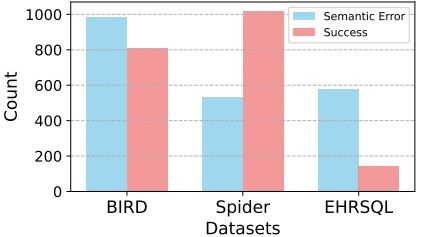

Figure 3: Visualization of class distribution of test sets across datasets.

From the Table 9 and the visualization in Figure 3, we observe a clear difference in the proportion of positive samples (incorrect SQL queries) across datasets: 54.76% for *BIRD*, 80.41% for *EHRSQL*, and only 34.30% for *Spider*. This indicates that *Spider* dataset contains not only fewer incorrect samples but also more subtle ones, making it harder for the model to accurately distinguish these rare and complex errors. As a result, in our experiments, *Spider* tends to show lower AUPRC compared to AUROC, while *BIRD* and *EHRSQL* having a higher proportion of incorrect samples are more likely to achieve higher AUPRC.

## E.2 EXPERIMENT CONFIGURATION

All experiments were conducted in a Python 3.12 environment. For neural network training and inference, we utilized PyTorch 2.8.0, PyTorch Geometric 2.6.1, Pytorch Scatter 2.1.2, Transformers 4.56.1, and Numpy 1.26.4 SQL preprocessing was performed using SQLGlot 27.7.0 to assist. And the CUDA version of our machine is 12.4. To generate the logical plans for SQL queries, we selected Apache Calcite (Begoli et al., 2018) as the query optimizer due to its extensive support for SQL parsing and optimization. This component was implemented in Java with JDK 21.0.8, utilizing Apache Calcite 1.40.0 as the core optimization library, sqlite-jdbc 3.50.3.0 for database connectivity, and Spring Boot 3.5.4 to enable web service integration and testing with Python environments, with all dependencies managed via Maven. Finally, all LLM-based baselines are accelerated with vLLM with version 0.9.1 during generation process. Please refer to our repository and che Our computational environment consists of a server equipped with 8 NVIDIA A100 GPUs, each with 80GB memory, ensuring sufficient resources for large-scale model training and experimentation. To enhance reproducibility, we fix random seeds with 2025 for all relevant libraries in each experiments.

We implement and evaluate each approach using the same experimental pipeline for a fair comparison. Unless otherwise specified, model parameters are optimized by the $\mathrm{AdamW}$ optimizer (Loshchilov & Hutter, 2017), with an initial learning rate of $1e\text{-}4$ and weight decay of $1e\text{-}4$. The training batch size is set to 32, and dropout is applied with a rate of $0.3$ to mitigate overfitting. Furthermore, an early-stopping strategy is employed by monitoring the validation AUROC metric, with a patience threshold of 5 epochs. For model selection, the best checkpoint is chosen based on the highest AUROC achieved in validation dataset during training.

## E.3 EVALUATION METRICS FOR TEXT-TO-SQL SEMANTIC VALIDATION

In evaluating our Text-to-SQL semantic validation task, a binary classification problem that predicts the correctness of SQL predictions, we adopt two widely used metrics: *Area Under the Precision-Recall Curve* (AUPRC) and *Area Under the Receiver Operating Characteristic Curve* (AUROC).

AUPRC measures the area under the curve that plots precision against recall for different threshold settings. This metric is particularly informative for imbalanced datasets, as it focuses on the model's ability to correctly identify positive examples and ignore the numerous negatives. In our context, a higher AUPRC indicates a better trade-off between precision and recall when distinguishing correct from incorrect SQL queries.

AUROC, on the other hand, represents the area under the ROC curve, which plots the true positive rate (sensitivity) against the false positive rate (1-specificity) across thresholds. AUROC provides an aggregate measure of performance across all classification thresholds, indicating the probability that the model ranks a randomly chosen positive instance higher than a negative one.

Both AUPRC and AUROC give us a comprehensive view of model's discriminative ability. As our primary objective is to detect semantic errors during the Text-to-SQL process, we **define invalid question-SQL pairs as positive samples, while valid pairs are treated as negative samples**.

## E.4 SENSITIVITY ANALYSIS OF QUERY OPTIMIZER FOR LOCIAL PLAN GENERATION

HEROSQL relies on logical plans (LPs) and abstract syntax trees (ASTs) as core foundational components. Notably, the method exhibits **weak sensitivity to the specific choice of query optimizer (aka. LP generator) or its optimization configurations**. Modern SQL query optimizers are inherently designed to retain the original query semantics, with their differences primarily manifesting in local decision-making processes—such as join order selection, predicate pushdown strategies, or the insertion of intermediate operators. These variations only induce minor structural discrepancies in the generated LPs and do not alter the high-level semantic information that underpins the validation mechanism of our framework.

Regarding the handling of logical plan extraction failures, these cases should not be considered as semantically incorrect. Instances where an LP cannot be successfully generated are almost exclusively attributed to the SQL query failing to pass the database compiler (e.g., due to syntax errors, invalid table/column references, or malformed clauses). In such scenarios, the optimizer abstains

from producing an LP entirely, and the database engine returns a clear compilation error. We categorize these cases as **syntax-level errors rather than semantic-level issues**. As depicted in Figure 1, the error messages generated during compilation are fed back into the iterative query regeneration pipeline with the large language model (LLM), which refines the SQL until it compiles successfully. Only after passing this compilation check does the query proceed to the semantic validation phase of HEROSQL.

## F  ALGORITHMS FOR HEROSQL

In this section, we detail the algorithm of context guided property embedding in the hierarchical intermediate representation of the SQL query in HEROSQL, which is shown in Algorithm 1. Besides, we also provide the workflow of the nested message passing process in Algorithm 2.

---

**Algorithm 1** Hierarchical SQL Representation of HEROSQL

---

**Require:** Training dataset $\mathcal{D}_{\text{train}} = \{(q_i, s_i)\}$, LLM-based embedding model $g$
1: **for** each logical plan node $v_i^{\mathcal{L}} \in V_{\mathcal{L}}$ **do**
2:     **for** each AST node $v_j^{\mathcal{A}_i} \in V_i$ of $\mathcal{A}_i$ **do**
3:         Fetch AST node embedding with guided context:

$$X = [\text{CONTEXT}; v_j^{\mathcal{A}_i}]$$

4:         Compute token embeddings $H = g(X)$
5:     **end for**
6: **end for**
7: **return** hierarchical intermediate representation of SQL query $s_i$

---

---

**Algorithm 2** Nested Message Passing over AST and LP

---

**Require:** Hierarchical Intermediate Representation of SQL query, AST steps $T_{\text{ast}}$, LP steps $T_{\text{logic}}$
1: **for** each logical plan node $v_i^{\mathcal{L}} \in V_{\mathcal{L}}$ **do**
2:     **for** each AST node $v_j^{\mathcal{A}_i} \in V_i$ of $\mathcal{A}_i$ **do**
3:         **for** $t = 1$ to $T_{\text{ast}}$ **do**
4:             $h_{v_j^{\mathcal{A}_i}}^{(t+1)} \leftarrow \text{UPDATE}_{\text{ast}}\left(h_{v_j^{\mathcal{A}_i}}^{(t)}, \text{AGGREGATE}_{\text{ast}}(\{h_{v_k^{\mathcal{A}_i}}^{(t)} : v_k^{\mathcal{A}_i} \in \mathcal{N}(v_j^{\mathcal{A}_i})\})\right)$
5:         **end for**
6:     **end for**
7:     $h_{v_i^{\mathcal{L}}} \leftarrow \text{POOLING}(\{h_{v_j^{\mathcal{A}_i}}^{(T_{\text{ast}})}\}_{v_j^{\mathcal{A}_i} \in V_i})$
8: **end for**
9: **for** $t = 1$ to $T_{\text{logic}}$ **do**
10:     **for** each logical plan node $v_i^{\mathcal{L}} \in V_{\mathcal{L}}$ **do**
11:         $h_{v_i^{\mathcal{L}}}^{(t+1)} \leftarrow \text{UPDATE}_{\text{logic}}\left(h_{v_i^{\mathcal{L}}}^{(t)}, \text{AGGREGATE}_{\text{logic}}(\{h_{v_k^{\mathcal{L}}}^{(t)} : v_k^{\mathcal{L}} \in \mathcal{N}_L(v_i^{\mathcal{L}})\})\right)$
12:     **end for**
13: **end for**
14: $h_{\text{SQL}} \leftarrow \text{POOLING}(\{h_{v_i^{\mathcal{L}}}^{(T_{\text{logic}})}\}_{v_i^{\mathcal{L}} \in V_{\mathcal{L}}})$
15: **return** embedding of SQL query $h_{\text{SQL}}$

---

## G  SENSITIVITY ANALYSIS

### G.1  IMPACT OF NUMBER OF AST-BASED AUGMENTED SAMPLES

To investigate how performance scales with the number of semantically incorrect samples generated through AST-driven Sub-SQL augmentation, we conducted experiments using HEROSQL with the *embeddinggemma-300m* backbone, varying the number of AST-based augmentation samples included in the training set. Specifically, we trained models under different negative-to-positive (N/P)

ratios: 0.55 (no AST-based augmentation), 0.8, 1.0, and 1.2, and evaluated performance on the *BIRD*, Spider, and *EHRSQL* datasets. The results are shown in Table 10.

Our findings show that increasing the N/P ratio from $0.55 \rightarrow 0.8 \rightarrow 1.0$ consistently improves performance for most datasets. The best results for *Spider* and *EHRSQL* appear at an N/P ratio of 1.0, while *BIRD*'s best is at 0.8, with only a marginal difference at 1.0, which corresponds to prior research's conclusions (Robinson et al., 2020; Gao et al., 2021; Wu et al., 2022; Chawla et al., 2002).

Table 10: Sensitivity analysis of different negative-to-positive (N/P) ratios for training dataset.

| N/P Ratio | BIRD | | Spider | | EHRSQL | |
|---|---|---|---|---|---|---|
| | AUPRC | AUROC | AUPRC | AUROC | AUPRC | AUROC |
| 0.55 | 64.39 | 60.56 | 49.73 | 66.19 | 84.14 | 60.77 |
| 0.8 | **68.20** | **63.54** | 50.60 | 66.55 | 85.30 | 63.55 |
| 1.0 | 67.10 | 63.48 | **51.11** | **69.26** | **85.48** | **65.92** |
| 1.2 | 65.24 | 61.44 | 48.35 | 65.74 | 84.52 | 56.19 |

However, increasing the ratio to 1.2 leads to a noticeable performance drop across all datasets, suggesting that an excessive number of negative samples may introduce noise, reduce signal-to-noise ratio, or skew the data distribution in a way that impairs learning.

Overall, these results indicate that **maintaining a balanced and trade-off negative-to-positive ratio of approximately 1:1 is an effective and robust strategy**. This ratio provides sufficient semantic contrast without overwhelming the model with noisy or overly abundant negative samples, ultimately supporting better generalization and more stable learning dynamics.

### G.2 IMPACT OF LAYER NUMBERS IN NMPNN

We examine the impact of layer numbers in the NMPNN of HEROSQL. We specifically analyze the variations of AUPRC and AUROC for different layer numbers $l$ from the list $[1, 2, 3, 4]$ with the embedding model *embeddinggemma-300m*. As shown in Figure 4, the performance reaches its peak when the layer number is set to 2 across almost all evaluation metrics. When the layer number exceeds 2, the model's performance degrades, likely due to increased training difficulty and the over-smoothing effect, where node

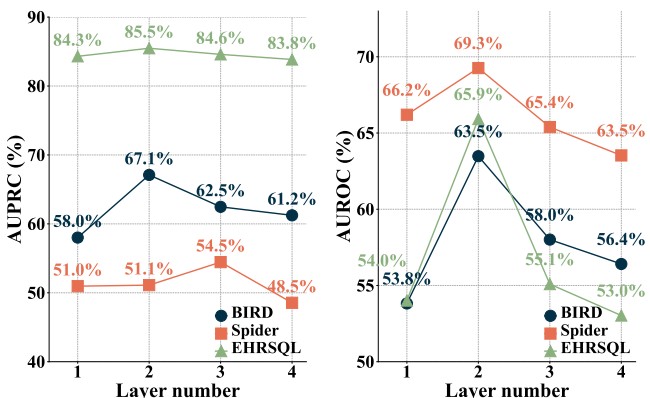

Figure 4: Sensitivity analysis of layer number $l$ in nested message passing neural network

representations become indistinguishable and fail to capture fine-grained structural details needed for effective semantic validation. Conversely, with only one layer, the model struggles to sufficiently aggregate structural information from both the LP and AST, which prevents it from fully modeling the relationships among different nodes in SQL queries. Therefore, a layer number of 2 provides the optimal balance between information propagation and feature discrimination.

## H ADDITIONAL EXPERIMENTS

### H.1 PERFORMANCE ON MORE CHALLENGING DATASET

To further verify the effectiveness of our method, we evaluate HEROSQL and baseline methods on more challenging datasets like *Spider 2.0* (Lei et al., 2024). *Spider 2.0* dataset is a quite challenging dataset whose questions are very difficult, schemas are much longer than other datasets, and requires complex SQL queries that may including nested SQL clauses, common table expression (CTE), and window functions to deal with them.

The results are shown shown in Table 1, HEROSQL achieves the best performance across all configurations on *Spider 2.0* dataset, outperforming all baselines and demonstrating strong robustness in handling complex SQL queries. These empirical results provide clear evidence that HEROSQL scales effectively to more complex and realistic SQL scenarios.

Actually, HEROSQL is designed with **sufficient generality to process any syntactically correct SQL statement** that can be compiled by the corresponding SQL engine, including those containing advanced constructs. As long as the SQL statement is syntactically valid, we can obtain its logical plan (LP) and abstract syntax tree (AST), upon which HEROSQL performs semantic validation. In Table 11, we present a complex SQL query with CTE Structures and show that HEROSQL can handle it well and reflect the error part.

Table 11: Case Study on Text-to-SQL Error Detection with HEROSQL in Handling Complex SQL with CTE Structures

| | |
|---|---|
| **Description** | HEROSQL is designed with sufficient generality to process any syntactically correct SQL statement that can be compiled by the underlying SQL engine, including advanced constructs such as nested queries, common table expressions (CTEs), and window functions. As long as a query is syntactically valid, the system retrieves its logical plan (LP) and abstract syntax tree (AST), upon which HEROSQL performs semantic validation. |
| **Example SQL** | [Question]
List the employees in the company whose salaries are higher than 100,000, as well as the average salary of their respective departments.

[SQL]
WITH DeptAvg AS (
SELECT Department, AVG(Salary) AS AvgSalary
FROM Employees
GROUP BY Department
),
HighEarners AS (
SELECT Name, Department, Salary
FROM Employees
WHERE Salary > 100000
)
SELECT h.Name, h.Salary, d.AvgSalary
FROM HighEarners h
JOIN DeptAvg d ON h.Department = d.Department; |
| **Logical Plan** | Project(Name=[0], Salary=[2], AvgSalary=[5])
Join(condition=[=(1, 4)], joinType=[inner])
Project(Name=[0], Department=[1], Salary=[2])
Filter(condition=[$\geq$(2, 100000)])
TableScan(table=[[Employees]])
Aggregate(group=[{Department}], AvgSalary=[AVG(2)])
Project(Department=[1], Salary=[2])
TableScan(table=[[Employees]]) |
| **Explanation** | The logical plan decomposes the two CTEs into standard relational operators, explicitly exposing the global semantics of the query. The first branch filters employees with salaries above 100,000, while the second computes the average salary for each department. The final join aligns high earners with the corresponding departmental averages. Operating on this structured LP/AST representation, HEROSQL robustly handles compositional and deeply nested SQL queries.

More broadly, logical-plan–based reasoning enables HEROSQL to capture the global intent and hierarchical organization inherent in sophisticated SQL. This design supports principled interpretation of nested SQL, CTE expansions, and window-function semantics, which encode analytical operations that are difficult to interpret reliably at the surface-syntax level. |

## H.2 Performance on Ambiguous Text-to-SQL Dataset

To evaluate the robustness of HEROSQL in real-world scenarios, we conducted experiments on the *Ambrosia* (Saparina & Lapata, 2024) dataset. *Ambrosia* contains naturally ambiguous questions, each paired with 2–3 correct SQL interpretations, designed to test the generalization ability of models under ambiguity and complex schema information. The experimental results on *Ambrosia* dataset are reported in Table 12.

Table 12: Performance comparison of different methods on the *Ambrosia* benchmark.

| Method | Qwen3-0.6b | | Gemma-3-0.3b | |
|---|---|---|---|---|
| | AUPRC | AUROC | AUPRC | AUROC |
| prompt | 58.18 | 48.75 | 58.36 | 50.00 |
| CoT | 59.04 | 49.79 | 58.38 | 50.04 |
| ConfScore | 59.61 | 50.63 | 59.58 | 51.56 |
| COVE | 57.88 | 48.80 | 58.64 | 50.58 |
| TED | 63.91 | 56.95 | 60.51 | 50.70 |
| HEROSQL | **64.18** | **57.28** | **61.72** | **52.06** |

Across all settings, HEROSQL maintains strong generalization ability on ambiguous and noisy-schema queries. These results indicate that HEROSQL is resilient to noise and schema-level ambiguity. This suggests that HEROSQL can generalize well even in real-world environments where schema noise and ambiguity are common.

## H.3 Latency and Throughput for Interactive Validation

To understand the computational cost of our method, we conduct experiments measuring end-to-end inference latency and throughput, comparing our method HEROSQL with the pure LLM baseline Prompt, which is the lightest method among all baselines. The test are conducted on 4 RTX 3090 GPUs using *Qwen3-0.6B* as the LLM backbone on BIRD dataset. The results are summarized in Table 13.

Table 13: Comparison of throughput and average latency between pure LLM method and HEROSQL.

| Method | Throughput (samples/s) | Avg. Latency (ms/sample) |
|---|---|---|
| Prompt | 130.21 | 7.68 |
| HEROSQL | 85.47 | 11.70 |

From the result in Table 13, we can see that **our approach maintains comparable throughput and latency to the Prompt-based method**, without introducing significant delays or reducing overall performance. This is because that the efficiency of our approach mainly stems from the lightweight two-layer *Nested Message Passing Neural Network* (NMPNN) backbone, which consists of GAT layers and small MLPs. The average per-sample latency (11.70 ms) is just slightly higher than that of the baseline (7.68 ms).

Overall, these results demonstrate that **HEROSQL provides competitive latency and throughput for interactive usage**. Having an average end-to-end per-query latency of 11.7 ms remains well within typical real-time validation constraints.

On scalability, we fully recognize the importance of handling long contexts and complex schemas in real-world enterprise scenarios. Our framework is deliberately designed to be model-agnostic and compatible with any scalable backbone LLM and message-passing architecture. This allows HEROSQL to naturally benefit from advances in scalable GNNs and graph training techniques, such as LMC (Shi et al., 2023), Sketch-GNN (Ding et al., 2022), REST (Xue et al., 2025) MeGraph (Dong et al., 2023), and Ginex (Park et al., 2022), as well as from future improvements in large-context LLMs.

For long-context scalability specifically, by adopting a decoder-only pretrained LLM as the embedding backbone, our method inherits the context window and scalability properties of state-of-the-art LLMs. This enables us to process very large and complex database schemas and SQL queries, on par with or beyond existing LLM-based Text-to-SQL validators.

## H.4 Performance Comparison Between HeroSQL and Commercial Models

To evaluate the performance difference between our approach and directly using prompts with commercial models for SQL semantic validation, we conducted experiments with strong commercial LLM backbones, including *GPT-4o*, *GPT-o4-mini*, and *Claude-haiku-4-5*. These models are used in a prompt-only setting, where the LLM directly judges SQL semantic correctness without any additional training.

For HeroSQL, we further evaluated different Qwen3 backbones, including Qwen3-0.6B and Qwen3-4B, to examine whether HeroSQL continues to provide improvements when paired with larger open-source models. The results are summarized in Table table 14.

Table 14: Comparison of the performance of different models.

| Method | Base | BIRD | | Spider | | EHRSQL | |
|---|---|---|---|---|---|---|---|
| | | AUPRC | AUROC | AUPRC | AUROC | AUPRC | AUROC |
| Prompt | Qwen3-0.6b | 60.85 | 57.25 | 40.86 | 58.94 | 84.72 | 58.86 |
| | GPT-4o | 73.56 | 73.45 | 47.68 | 66.37 | 86.78 | 67.04 |
| | GPT-o4-mini | **80.36** | **79.25** | **63.21** | **75.81** | 86.38 | 68.00 |
| | Claude-haiku-4-5 | 75.87 | 76.15 | 48.72 | 65.80 | 83.92 | 61.26 |
| HeroSQL | Qwen3-0.6b | 67.39 | 61.51 | 51.92 | 67.32 | 89.07 | 69.53 |
| | Qwen3-4b | 71.18 | 64.17 | 53.52 | 69.78 | **89.28** | **72.03** |

From these results, we have the following observations. ❶ **Model capability correlates with prompt-only baseline performance**: stronger commercial LLMs (e.g., GPT-o4-mini, GPT-4o, Claude-haiku-4-5) outperform lightweight open-source prompt baselines on all datasets. ❷ **Within the Qwen family, scaling up improves performance**: HeroSQL with *Qwen3-4B* consistently outperforms HeroSQL with *Qwen3-0.6B*, confirming that our framework naturally benefits from more capable backbones. ❸ HeroSQL adds value on top of a given backbone: even when starting from a relatively small open-source model *Qwen3-0.6B*, HeroSQL achieves substantial gains over its prompt-only counterpart, and with Qwen3-4B it reaches performance that is competitive with or better than strong commercial prompt-only baselines on several metrics.

It is worth noting that our contribution is a framework for hierarchical SQL semantic validation, rather than a new pretrained model. HeroSQL is model-agnostic: its capability depends on the underlying pretrained LLM-based embedding model and is compatible with a wide range of pretrained models, from lightweight open-source models to state-of-the-art commercial LLMs.

The main experiments are conducted primarily with two small open-source encoders *Qwen3-0.6B* and *Gemma-3-0.3B* to demonstrate that the gains come from the framework itself, not merely from model scale.

When selecting LLM backbones, we chose open-source models instead of commercial ones mainly for three practical reasons:

- **Privacy**. Real-world database applications often involve sensitive or proprietary data. In many enterprise settings, only local offline deployment is allowed, and sending data to external commercial APIs to call these commercial but powerful models is not acceptable.

- **Cost**. Training HeroSQL requires repeatedly invoking embedding models over large augmented datasets. Using commercial APIs as embedding backbones would incur prohibitive costs.

- **Efficiency**. Frequent remote API calls for embeddings would also significantly slow down training and experimentation, making the framework impractical for real deployment and iteration.

## I Data Ethics Statement

To evaluate the efficacy of our work, we conducted experiments using 5 datasets, including *BIRD*, *Spider*, *EHRSQL*, *Amrosia*, and *Spider 2.0*. All datasets are publicly available and used in accordance with their respective terms of use. No personally identifiable information was involved, and no human or animal subjects participated in this research.

## J  THE USE OF LARGE LANGUAGE MODELS

In this work, Large Language Models (LLMs) are utilized in polish writing and code development. In particular, LLMs helped refine the language and clarity of the paper, such as enhancing grammar and improving stylistic quality. LLMs were also employed to support the development of experimental code, including providing coding suggestions and troubleshooting assistance. All LLM-generated content was thoroughly reviewed and verified by the authors prior to inclusion. Research design, critical analyses, and all final decisions were carried out independently by the authors.

## K  GUIDELINE FOR REVIEWERS

In the revised manuscript, different font colors are used to highlight the modifications made in response to each reviewer's comments, as detailed below.

- Reviewer LazQ: ● and ●
- Reviewer Z8MB: ●, ● and ●
- Reviewer mSpA: ●, ●, and ●
- Reviewer 3mDL: ●, ●

