# OpenReview forum: "Bridging Global Intent with Local Details: A Hierarchical Representation Approach for Semantic Validation in Text-to-SQL"
_ICLR.cc/2026/Conference — Submitted to ICLR 2026_

### Official Review · Reviewer_3mDL · 2025-10-26

**Soundness:** 2
**Presentation:** 1
**Contribution:** 3
**Rating:** 2
**Confidence:** 4

**Summary:**

This paper addresses the problem of semantic validation in Text-to-SQL systems, determining whether a generated SQL query correctly captures the meaning of a natural language question. The authors propose HEROSQL, a hierarchical representation approach that integrates Logical Plans (LP) for global semantic intent and Abstract Syntax Trees (AST) for local structural details. These representations are processed via a Nested Message Passing Neural Network (NMPNN) to capture multi-level semantic relations. To overcome data scarcity, the paper introduces an AST-based sub-SQL augmentation strategy that generates challenging negative samples, supplemented by LLM-based data augmentation. Experiments on BIRD, Spider, and EHRSQL datasets show improvements in AUPRC and AUROC over several baselines (Prompting, CoT, ConfScore, COVE, TED).

**Strengths:**

- *S1- Novel hierarchical representation*: Integrating LP and AST for semantic validation is a promising and technically meaningful idea. It captures both global intent and local structural details.

- *S2- Empirical gains*: The method achieves consistent improvements over existing baselines across multiple datasets, suggesting its effectiveness in capturing fine-grained semantic errors.

- *S3- Augmentation strategy*: The AST-driven negative data generation is a creative way to address data scarcity for supervised semantic validation.

**Weaknesses:**

- *W1- Clarity and organization*:
  The paper is difficult to follow. While individual components (LP, AST, NMPNN) are described in depth, their integration and flow within the full pipeline are not clearly presented. Figure 1 is dense and does not sufficiently illustrate how the components interact. A concise system overview early in the paper would help orient readers.

- *W2- Baseline clarity and fairness*:
  The baselines (Prompt, CoT, ConfScore, COVE, TED) are only briefly mentioned, and their adaptation to the semantic validation setting is unclear. For instance, Chain-of-Thought (Wei et al., 2022) and CoVe (Dhuliawala et al., 2024) are general approaches; the paper should explain how they are adapted for text-to-SQL validation. Moreover, HEROSQL is trained in a fully supervised manner using labeled (question, SQL) pairs, while most baselines appear unsupervised or heuristic. This difference should be acknowledged and controlled for (e.g., through fine-tuned supervised baselines).

- *W3- Assumptions not fully justified*:
  The claim that hierarchical intermediate representations yield more accurate semantics is plausible but not empirically validated. The assumption that LLM- and AST-based augmentations generalize well remains underexplored. The out-of-domain test on EHRSQL is too limited (only two databases) to conclusively demonstrate generalization.

- *W4- Writing and presentation quality*:
  The prose is verbose, with long sentences and repeated statements. Important contributions and intuitions are buried in technical detail. The introduction and methodology sections would benefit from clear motivation, schematic diagrams, and more intuitive examples of semantic errors being detected.

**Questions:**

1. How are the unsupervised baselines adapted to the supervised validation setting? Are they fine-tuned on the same augmented data?

2. Can you provide qualitative examples of semantic inconsistencies that HEROSQL detects but baseline methods miss? I can see some examples in the appendix but I cannot tell how the baselines perform on them.

3. Have you analyzed how performance scales with the number or diversity of negative samples generated by the AST-based augmentation?

---

> ### Author Response · Authors · 2025-11-22
> **Response to [W1 & W4 – Clarity, Writing, and Presentation Quality]**
>
> We thank the reviewers (W1 and W4) for their constructive comments on clarity and presentation. We have carefully revised the manuscript accordingly.
>
> - To improve readability, we have **polished the writing in both the Introduction and Method sections**. In particular, we shortened long sentences, removed redundant phrasing, and reorganized some paragraphs so that the main ideas and contributions are easier to follow. The revised manuscript has been updated in OpenReview.
>
> - To better present our research motivation, **we have updated Figure 1**. The new figure explicitly illustrates the difference between syntactic error detection and semantic detection, and **highlights the motivation for our work in addressing semantic-level issues**.
>
> - To clarify the overall pipeline and the roles of LP, AST, and NMPNN, we have **updated the previous method figure (originally Figure 1, now Figure 2)**. The new version is a more explicit process **diagram that visualizes the workflow and the information flow among LP, AST, and NMPNN, making the relationships between these components clearer to the readers.**

---

> ### Author Response · Authors · 2025-11-22
> **Response to [W2: Baseline Clarity and Fairness]**
>
> Thank you for your valuable comments. We would like to clarify that the detailed adaptations of all baseline methods (Prompt, CoT, ConfScore, COVE, TED) **are implemented under rigorous fairness settings, and the corresponding descriptions are provided in Appendix E.1**.
>
> - In particular, Appendix E.1 explains **how each method is instantiated in our setting**, addressing concerns about how these approaches are applied and ensuring that the comparisons are task-appropriate. In the revised manuscript, we have further expanded these descriptions to improve transparency and reproducibility.
>
> - Regarding supervision and fairness, while HEROSQL is trained in a fully supervised manner on labeled (question, SQL) pairs, the TED baseline is also a fully supervised method specifically designed for SQL validation. To **ensure a fair comparison**, we additionally conduct ablation studies where HEROSQL is trained under comparable fully supervised settings without AST guidance, and without both AST and LP, as reported in Table 1 in Section 5. These results show that (i) the AST and LP components are crucial for the performance gains of HEROSQL, and (ii) HEROSQL consistently outperforms all baselines, including TED, under matched supervision levels.

---

> ### Author Response · Authors · 2025-11-22
> **Response to [W3: Assumptions not fully justified]**
>
> Thank you for your insightful comments regarding the justification of our assumptions and the evaluation of generalization.
>
> To address your concerns about generalization, it is important to clarify the sources of stochasticity in our approach. **In HEROSQL, the use of ASTs and LPs does not introduce randomness**. **ASTs are deterministic, abstract structures that capture hierarchical information within SQL queries, and LPs are deterministic representations of execution semantics**. Their consistent structural representations transcend the idiosyncrasies of specific SQL dialects and domain knowledge. This is why numerous prior works (e.g., [1, 2, 3]) have leveraged AST-/plan-centric modeling for robust generalization across databases and tasks. This literature supports our assertion that these structured representations provide domain-agnostic abstractions that facilitate transferability and generalization.
>
> The principal source of variability in our method stems from the use of pretrained LLMs, which we employ to generate embeddings. Since these LLMs have been pretrained on diverse and large-scale corpora, **their representations exhibit substantial generalization capability**. This generalization ability is reflected both conceptually and empirically in our results: as shown in Table 1, incorporating **LP and AST information consistently enhances performance across all evaluated datasets.**
>
> Moreover, in response to your concern regarding the scope of out-of-domain evaluation, we have extended our experiments to **include two additional datasets, Ambrosia [4] and Spider2 [5]**. The results are summarized below:
>
> | Method    | Ambrosia (Qwen3-0.6B AUPRC) | Ambrosia (Qwen3-0.6B AUROC) | Ambrosia (Gemma-3-0.3B AUPRC) | Ambrosia (Gemma-3-0.3B AUROC) | Spider2 (Qwen3-0.6B AUPRC) | Spider2 (Qwen3-0.6B AUROC) | Spider2 (Gemma-3-0.3B AUPRC) | Spider2 (Gemma-3-0.3B AUROC) |
> |----------|------------------------------|------------------------------|--------------------------------|--------------------------------|-----------------------------|------------------------------|--------------------------------|--------------------------------|
> | Prompt   | 58.18                        | 48.75                        | 58.36                          | 50.00                          | 88.78                       | 51.80                        | 88.30                          | 44.58                          |
> | CoT      | 59.04                        | 49.79                        | 58.38                          | 50.04                          | 89.15                       | 48.32                        | 89.22                          | 50.00                          |
> | ConfScore| 59.61                        | 50.63                        | 59.58                          | 51.56                          | 90.15                       | 44.77                        | 90.79                          | 52.34                          |
> | COVE     | 57.88                        | 48.80                        | 58.64                          | 50.58                          | 90.75                       | 61.86                        | 88.51                          | 45.86                          |
> | TED      | 63.91                        | 56.95                        | 60.51                          | 50.70                          | 91.44                       | 58.06                        | 89.14                          | 49.56                          |
> | HEROSQL  | **64.18**                    | **57.28**                    | **61.72**                      | **52.06**                      | **92.59**                   | **64.32**                    | **91.55**                      | **59.85**                      |
>
> Neither Ambrosia nor Spider2 is used during training, **so they provide genuine out-of-domain evaluation**. Notably, HEROSQL achieves the best performance on both datasets, further demonstrating the robustness and generalization potential of our approach beyond the EHRSQL test set.
>
> > [1] A Comparative Study and Component Analysis of Query Plan Representation Techniques in ML4DB Studies, VLDB 2024
> > [2] SyntaxSQLNet: Syntax Tree Networks for Complex and Cross-Domain Text-to-SQL Task, EMNLP 2018
> > [3] LearNAT: Learning NL2SQL with AST-guided Task Decomposition for Large Language Models, arXiv 2025
> > [4] AMBROSIA: A Benchmark for Parsing Ambiguous Questions into Database Queries, NeurIPS 2024
> > [5] Spider 2.0: Evaluating Language Models on Real-World Enterprise Text-to-SQL Workflows, ICLR 2025

---

> ### Author Response · Authors · 2025-11-22
> **Response to [Q1: How are the unsupervised baselines adapted to the supervised validation setting? Are they fine-tuned on the same augmented data?]**
>
> - For the baselines that do not require supervised training—namely Prompt, CoT, ConfScore, and COVE—we leverage the **inherent generalisation capabilities of LLMs and design task-specific prompts that adapt these methods to the Text-to-SQL semantic validation setting**. These prompts are carefully crafted to align model behaviour with the goals of semantic validation while preserving the unsupervised nature of the methods. The detailed adaptation strategies for each baseline are provided in Appendix E.1.
>
> - In contrast, for methods that rely on supervised training, such as TED and HEROSQL, **we adopt the same training conditions for fairness**. Both TED and HEROSQL are fine-tuned on the augmented data under identical experimental settings. This ensures a consistent and equitable comparison across all baselines, regardless of whether they are supervised or unsupervised.

---

> ### Author Response · Authors · 2025-11-23
> **Response to [Q2: Can you provide qualitative examples of semantic inconsistencies that HEROSQL detects but baseline methods miss? ]**
>
> Yes, let's consider the following two examples that errors can be detected by HEROSQL but missed by baseline.
>
> #### Case 1
>
> ```
> [Question]
> What is the average score in writing for the schools that were opened after 1991 or closed before 2000? List the school names along with the score. Also, list the communication number of the schools if there is any. Communication number refers to phone number.
>
> [Predicted SQL]
> SELECT s.School, s.Phone, satscores.AvgScrWrite
>     FROM schools s
>         JOIN satscores
>         ON s.CDSCode = satscores.cds
>     WHERE s.OpenDate > '1991-12-31'
>         OR s.ClosedDate < '2000-01-01'
>
> [Ground Truth SQL]
> SELECT s.School, s.Phone, satscores.AvgScrWrite
>     FROM schools AS s
>         LEFT JOIN satscores
>         ON s.CDSCode = satscores.cds
>     WHERE s.OpenDate > '1991-12-31'
>         OR s.ClosedDate < '2000-01-01'
> ```
>
> The error in this case **is difficult to detect even for human experts**.  "Average score" described in the question does not mean directly taking out the existing AvgScrWrite column, but rather performing an AVG aggregation again on the writing scores that meet the criteria.
> This case lacks an additional average aggregation step for the eligible writing scores. This omission is difficult to detect for baselines like TED that only focus on local details. Even baseline methods adept at global information understanding, including Prompt, CoT, and ConfScore, fail to identify this error.
> However, for HEROSQL, the absence of an aggregation node in the Logical Plan leads to a significant semantic deviation—one that allows HEROSQL to detect the error with ease.
>
> #### Case 2
>
> ```
> [Question]
> How many schools in Fresno (directly funded) have number of test takers not more than 250?
>
> [Predicted SQL]
> SELECT COUNT(*)
>     FROM schools
>         JOIN satscores
>         ON schools.CDSCode = satscores.cds
>     WHERE schools.City = 'Fres'
>         AND schools.FundingType = 'Directly Funded'
>         AND satscores.NumTstTakr <= 250;
>
> [Ground Truth SQL]
> SELECT COUNT(*)
>     FROM schools
>         JOIN satscores
>         ON schools.CDSCode = satscores.cds
>     WHERE schools.City = 'Fresno'
>         AND schools.FundingType = 'Directly Funded'
>         AND satscores.NumTstTakr <= 250;
> ```
>
> In this cass, the LLM made a small mistake: it wrote the filtering condition for schools.City as "Fres" instead of "Fresno."
> For most baselines that only focus on global intent while ignoring local details (such as Prompt, CoT, ConfScore, and COVE), this kind of detail error tends to be overlooked, so they fail to discover that "Fresno" was mistakenly written as "Fres." In contrast, the attention to local details guided by the AST in HEROSQL enables it to identify this detail error.

---

> ### Author Response · Authors · 2025-11-23
> **Response to [Q3: Have you analyzed how performance scales with the number or diversity of negative samples generated by the AST-based augmentation?]**
>
> Thank you for your valuable suggestion regarding the analysis of how performance scales with the number or diversity of negative samples generated through AST-based augmentation. To address this, **we conducted experiments using HEROSQL with the Gemma-3-0.3B backbone, varying the number of AST-generated negative samples included in the training set.**
>
> Specifically, we trained models under different negative-to-positive (N/P) ratios: **0.55** (no AST-based augmentation), **0.8**, **1.0**, and **1.2**, and evaluated performance on the BIRD, Spider, and EHRSQL datasets. The results are shown below:
>
> | N/P Ratio | BIRD AUPRC | BIRD AUROC | Spider AUPRC | Spider AUROC | EHRSQL AUPRC | EHRSQL AUROC |
> |-----------|------------|-------------|--------------|--------------|--------------|---------------|
> | 0.55      | 64.39      | 60.56       | 49.73        | 66.19        | 84.14        | 60.77         |
> | 0.8       | **68.20**  | **63.54**   | 50.60        | 66.55        | 85.30        | 63.55         |
> | 1.0       | 67.10      | 63.48       | **51.11**    | **69.26**    | **85.48**    | **65.92**     |
> | 1.2       | 65.24      | 61.44       | 48.35        | 65.74        | 84.52        | 56.19         |
>
> Our findings show that increasing the N/P ratio from **0.55 → 0.8 → 1.0** consistently improves performance across most datasets. The best results for Spider and EHRSQL appear at an N/P ratio of **1.0**, while BIRD’s best is at **0.8**, with only a marginal difference at **1.0**, which corresponds to the works [1,2,3,4].
>
> However, increasing the ratio to **1.2** leads to a noticeable performance drop across all datasets, suggesting that an excessive number of negative samples may introduce noise, reduce signal-to-noise ratio, or skew the data distribution in a way that impairs learning.
>
> Overall, these results indicate that **maintaining a balanced and trade-off negative-to-positive ratio of approximately 1:1 is an effective and robust strategy**. This ratio provides sufficient semantic contrast without overwhelming the model with noisy or overly abundant negative samples, ultimately supporting better generalization and more stable learning dynamics.
>
> These new analyses and results have been incorporated into the revised manuscript for completeness and to fully address your question.
>
> > [1] Contrastive Learning with Hard Negative Samples, ICLR 2021.
> > [2] SimCSE: Simple Contrastive Learning of Sentence Embeddings,  EMNLP 2021.
> > [3] ESimCSE: Enhanced Unsupervised Contrastive Learning for Sentence Embedding, COLING 2022.
> > [4] SMOTE: Synthetic Minority Over-sampling Technique, JAIR 2002.

---

> ### Author Response · Authors · 2025-11-23
> **Summary of Responses to Reviewer 3mDL**
>
> We thank the reviewer for the careful and technically detailed feedback, which helped us substantially tighten both the theory and the presentation.
>
> - **W1 & W4 – Clarity, Writing, and Presentation Quality**:
>   We have substantially improved the clarity and presentation of the manuscript by polishing the writing in the Introduction and Method sections (shortening long sentences, removing redundancy, and reorganizing paragraphs). We also updated Figure 1 to better distinguish syntactic vs. semantic error detection, and redesigned the method figure (now Figure 2) to more clearly illustrate the overall pipeline and the roles of LP, AST, and NMPNN.
>
> - **W2: Baseline Clarity and Fairness**:
>   We clarified that all baselines (Prompt, CoT, ConfScore, COVE, TED) are implemented under rigorous fairness settings, with detailed adaptations documented in Appendix E.1. We further emphasized that both TED and HEROSQL are trained under matched fully supervised conditions, and our ablation studies (removing AST and removing both AST+LP) demonstrate that HEROSQL’s gains come from these components while still outperforming all baselines under comparable supervision.
>
> - **W3: Assumptions not fully justified**:
>   We explained that AST and LP are deterministic, domain-agnostic structural representations and do not introduce extra randomness, while the main stochasticity comes from pretrained LLMs with strong inherent generalisation. To strengthen our generalisation claims, we added out-of-domain experiments on Ambrosia and Spider2, where HEROSQL achieves the best performance across all baselines, further supporting its robustness beyond EHRSQL.
>
> - **Q1: How are the unsupervised baselines adapted to the supervised validation setting?**:
>   We clarified that unsupervised baselines (Prompt, CoT, ConfScore, COVE) are adapted via carefully designed task-specific prompts without fine-tuning, with details in Appendix E.1. For supervised methods (TED and HEROSQL), we use identical training settings and the same augmented data to ensure a fair and consistent comparison.
>
> - **Q2: Qualitative examples of semantic inconsistencies**:
>   We provided two concrete case studies showing errors that HEROSQL detects but baselines miss: (i) a missing aggregation despite an “average” requirement in the question (a global semantic mismatch revealed by the LP), and (ii) a subtle local error where “Fresno” is incorrectly written as “Fres” (captured via AST-level local inspection). These examples illustrate HEROSQL’s ability to jointly leverage LP (global semantics) and AST (local details).
>
> - **Q3: Scaling with number/diversity of negative samples**:
>   We added a detailed study varying the negative-to-positive (N/P) ratio (0.55, 0.8, 1.0, 1.2) using HEROSQL with Gemma-3-0.3B on BIRD, Spider, and EHRSQL. Results show improvements when increasing the N/P ratio up to around 1.0, and performance degradation when further increasing to 1.2, indicating that an excessive number of negatives can be harmful. We conclude that a roughly 1:1 N/P ratio is a robust and effective trade-off, consistent with prior work on contrastive learning and class balancing.
> ---
>
> In summary, we sincerely appreciate the reviewers’ constructive feedback. **We provide a “Guideline for Reviewers” in the appendix, where all major revisions are highlighted with colour to clearly indicate how each change corresponds to specific reviewer comments, enabling quick and convenient inspection.**

---

### Official Review · Reviewer_mSpA · 2025-11-01

**Soundness:** 3
**Presentation:** 3
**Contribution:** 3
**Rating:** 6
**Confidence:** 4

**Summary:**

HEROSQL addresses the challenge of semantic validation in Text-to-SQL, detecting when a syntactically valid query misaligns with user intent. This work introduces a hierarchical representation that fuses global intent (via logical plans) with local structure (via abstract syntax trees) and propagates information through a Nested Message Passing Neural Network. To overcome the scarcity of fine-grained labels, it employs an AST-driven sub-SQL augmentation strategy to generate diverse negative examples. In in-domain and out-of-domain benchmarks, it achieves an average +12.18 AUPRC and +13.41 AUROC improvement over prior methods.

**Strengths:**

1. This paper offers a clear and rigorous definition of semantic correctness in Text-to-SQL, grounded in the alignment between unstructured user intent and structured SQL components (at the AST level). While many prior works use the term loosely, this formalization is precise and actionable, and sets a strong foundation for future research in this area.

2. The technical design is solid and well-motivated. By combining AST-level and logical-plan encodings through a Nested Message Passing Neural Network, the model captures both local structure and global query intent, leading to rich, compositional SQL representations for validation.

3. HEROSQL consistently outperforms baselines across in-domain (BIRD, Spider) and out-of-domain (EHRSQL) datasets, using multiple small LLM backbones, and achieves substantial AUPRC and AUROC gains, demonstrating both effectiveness and generalizability.

**Weaknesses:**

1. AST perturbations are limited to the hand-crafted rule set (e.g. swapping operators, dropping predicates), which may not mimic the full spectrum of mistakes that LLMs actually make. LLM-generated negatives help diversify errors, but they’re still filtered purely by execution mismatch, so edge-case logic bugs that happen to produce the same result go unnoticed.

2. LLM-based baselines only include small LLMs. Could you justify this setup? What would be the performance if you use a commercial model like claude 4, gpt-4o, gpt-o4-mini? I suspect these models would be much stronger in determining SQL semantic correctness.

3. Improvements in AUROC/AUPRC show a better validator. However, how do the false positives and negatives in this validator affect the downstream application (e.g., text2sql)? For example, FP may lead to overcorrection and regress an originally correct SQL. In practice, if a user wants to use this validator in their text2sql pipeline, what threshold should they set? It would be good if the paper could show that this validator can really improve text2sql e2e accuracy through this error detection and correction step. Additionally, adding a discussion on how end users should use this validator (e.g., how to set the threshold) would make the paper stronger.

**Questions:**

1. What is the latency of the validator for determining semantic correctness?
2. Why only compare with small models? Could you compare with commercial LLMs such as claude 4?
3. AUPRC is much lower on Spider. Could you explain the reason?
4. In your results, is the positive class defined as correct or incorrect SQL? What is the exact class distribution in your training and evaluation datasets? Since class imbalance can significantly influence metrics like AUPRC and AUROC, clarifying this distribution is important for properly interpreting the results.
5. Could you show that your validator can improve the end-to-end performance of a text2sql pipeline? For example, use a very simple prompt to ask claude 4 do text2sql on BIRD dev, use your validator to detect semantic errors, and ask the LLM to correct erroneous queries.
6. Is it possible to use your method on benchmarks with more complicated SQL queries (e.g., spider 2.0)? It's ok that the answer is no. but would like to understand the potential challenges.

---

> ### Author Response · Authors · 2025-11-23
> **Response to [W1: AST perturbations are limited to the hand-crafted rule set and edge-case logic bugs that happen to produce the same result go unnoticed.]**
>
> Thank you for pointing out the limitations of our negative sample generation strategy.
>
> We fully acknowledge that generating negatives via AST perturbations is inherently constrained by a hand-crafted rule set and therefore cannot exhaustively cover the full spectrum of errors that LLMs may produce. Likewise, relying on execution mismatch as a filtering criterion may miss rare edge-case logic bugs that coincidentally yield the same result as the ground-truth query, leaving certain subtle errors undetected. However, we note that **designing negative augmentation schemes that fully address these issues remains an open problem**, and is an active research topic in the broader literature [1, 2, 3].
>
> In the context of HEROSQL, our primary focus is to **leverage hierarchical representations (LP and AST) to validate SQL semantics**, rather than to perfectly enumerate all possible error modes. To partially mitigate the limitations of rule-based AST perturbations, our training data **also includes negative examples produced directly by LLMs**, which broadens the diversity of observed error patterns and better approximates the mistakes that occur in real Text-to-SQL generation.
>
> Regarding logic bugs that are execution-equivalent to the gold query, we agree that these are intrinsically challenging. At the same time, such cases are generally **rare and largely beyond the scope of current evaluation protocols**. Mainstream Text-to-SQL benchmarks such as BIRD and Spider adopt execution-based metrics and treat queries as correct if their outputs match the gold results. Our methodology follows this widely used evaluation standard to ensure fair and consistent comparison with prior work.
>
> In summary, while we recognize the limitations of AST-based negative generation and execution-based filtering, our contribution is chiefly to advance semantic verification via hierarchical representations, and we have taken concrete, practical steps (including LLM-generated negatives) to diversify negative samples within the realistic constraints of this setting.
>
> > [1] Generating Enhanced Negatives for Training Language-Based Object Detectors, CVPR 2024
> > [2] RL on Incorrect Synthetic Data Scales the Efficiency of LLM Math Reasoning by Eight-Fold, NeurIPS 2025
> > [3] A Novel Negative Sample Generation Method for Contrastive Learning in Hierarchical Text Classification, COLING 2025

---

> ### Author Response · Authors · 2025-11-23
> **Response to [W2/Q2: Why do LLM-based baselines only include small LLMs? What would be the performance if you use a commercial model like claude 4, gpt-4o, gpt-o4-mini?]**
>
> Our contribution is a **framework for hierarchical SQL semantic validation**, rather than a new pretrained model. HEROSQL is **model-agnostic**: its capability depends on the underlying encoder LLM and is compatible with a wide range of pretrained models—from lightweight open-source models to state-of-the-art commercial LLMs. Our initial experiments primarily used small open-source encoders (e.g., Qwen3-0.6B and Gemma-3-0.3B) to demonstrate that the gains come from the framework itself, not merely from model scale.
>
> When selecting LLM backbones, we chose open-source models instead of commercial ones mainly for three practical reasons:
>
> - **Privacy.** Real-world database applications often involve sensitive or proprietary data. In many enterprise settings, only local, offline deployment is allowed, and sending data to external commercial APIs (e.g., GPT-4o, Claude) is not acceptable.
> - **Cost.** Training HEROSQL requires repeatedly invoking embedding models over large augmented datasets. Using commercial APIs as embedding backbones would incur prohibitive costs.
> - **Efficiency.** Frequent remote API calls for embeddings would also significantly slow down training and experimentation, making the framework impractical for real deployment and iteration.
>
> To address your concern more concretely, we have **added experiments with stronger LLM-prompt baselines**, including GPT-4o, GPT-o4-mini, and Claude-haiku-4-5. These models are used in a **prompt-only setting**, where the LLM directly judges SQL semantic correctness without any additional training.
>
> For HEROSQL, we further evaluated different Qwen3 backbones, including Qwen3-0.6B and Qwen3-4B, to examine whether HEROSQL continues to provide improvements when paired with larger open-source models. The results are summarized below:
>
> | Model                | BIRD AUPRC | BIRD AUROC | Spider AUPRC | Spider AUROC | EHRSQL AUPRC | EHRSQL AUROC |
> |----------------------|-----------:|-----------:|-------------:|-------------:|-------------:|-------------:|
> | GPT-4o               | 73.56      | 73.45      | 47.68        | 66.37        | 86.78        | 67.04        |
> | GPT-o4-mini          | **80.36**  | **79.25**  | **63.21**    | **75.81**    | 86.38        | 68.00        |
> | Claude-haiku-4-5     | 75.87      | 76.15      | 48.72        | 65.80        | 83.92        | 61.26        |
> | Qwen3-0.6B (prompt)  | 60.85      | 57.25      | 40.86        | 58.94        | 84.72        | 58.86        |
> | HEROSQL (Qwen3-0.6B) | 67.39      | 61.51      | 51.92        | 67.32        | 89.07        | 69.53        |
> | HEROSQL (Qwen3-4B)   | 71.18      | 64.17      | 53.52        | 69.78        | **89.28**    | **72.03**    |
>
> From these results, we observe:
>
> - **Model capability correlates with prompt-only baseline performance**: stronger commercial LLMs (e.g., GPT-o4-mini, GPT-4o, Claude-haiku-4-5) outperform lightweight open-source prompt baselines on most datasets.
> - **Within the Qwen family, scaling up improves performance**: HEROSQL with Qwen3-4B consistently outperforms HEROSQL with Qwen3-0.6B, confirming that our framework naturally benefits from more capable backbones.
> - **HEROSQL adds value on top of a given backbone**: even when starting from a relatively small open-source model (Qwen3-0.6B), HEROSQL achieves substantial gains over its prompt-only counterpart, and with Qwen3-4B it reaches performance that is competitive with or better than strong commercial prompt-only baselines on several metrics.
>
> Overall, these experiments support two key points:
> (1) better LLMs do improve raw semantic validation performance, and
> (2) **HEROSQL consistently provides additional benefits on top of the chosen encoder**, indicating that our hierarchical semantic validation framework is complementary to, rather than a substitute for, advances in LLM backbone quality.

---

> ### Author Response · Authors · 2025-11-23
> **Response to [W3: What threshold should they set?]**
>
> Thank you for raising this important point about how to choose a practical decision threshold for our validator in a Text-to-SQL pipeline.
>
> We agree that the threshold of the binary classifier should **depend on the requirements of the downstream application**. Improved AUROC/AUPRC means that our method yields a better Pareto frontier, allowing practitioners to select more favorable precision–recall trade-offs for different risk profiles.
>
> Concretely:
> - In scenarios where **false positives (FP)** are more harmful (e.g., over-correcting originally correct SQL and potentially regressing performance), one should choose **a higher threshold that prioritizes precision**, even at the cost of recall. In practice, this can be done by selecting the largest threshold at which precision stays above a desired target (e.g., ≥ 95%).
> - In scenarios where **false negatives (FN)** are more costly (e.g., it is critical not to miss any erroneous SQL being executed), one may prefer **a lower threshold that emphasizes high recall**, while allowing precision to vary within an acceptable range.
>
> In our experiments, we report results using the threshold that **maximizes the F1 score** on a validation set, providing a balanced operating point between precision and recall. This is a standard choice when there is no application-specific preference between FP and FN.
>
> In the revised manuscript, we will (i) add a brief discussion on threshold selection strategies tailored to different application needs, and (ii) further illustrate that integrating our validator with a reasonable threshold for error detection and targeted correction can improve end-to-end Text-to-SQL accuracy.

---

> ### Author Response · Authors · 2025-11-23
> **Response to [Q1: What is the latency of the validator for determining semantic correctness?]**
>
> To quantitatively assess the latency of our approach, we evaluated HEROSQL on the BIRD dataset and measured the average time required to verify the semantic correctness of each case. HEROSQL achieves an average validation latency of **11.70 ms per sample** and 85.47 throughput (samples/s), which is significantly lower than the typical time needed for SQL generation by LLMs.
>
> This lightweight overhead means that HEROSQL can be seamlessly integrated into real-time or interactive Text-to-SQL systems, providing semantic validation without introducing noticeable delays.

---

> ### Author Response · Authors · 2025-11-23
> **Response to [Q3: AUPRC is much lower on Spider. Could you explain the reason?]**
>
> The lower AUPRC observed on Spider is mainly due to **severe class imbalance in the test set**, where the proportion of positive (error) and negative (correct) samples strongly affects the magnitude of AUPRC (see also the detailed statistics in our response to Q4).
>
> Compared with datasets like BIRD, **Spider is relatively easier for LLMs**, leading to higher overall accuracy and thus **very few erroneous (positive) cases** in the test set. As a result:
>
> - The rare errors that do occur on Spider tend to be **extremely subtle, fine-grained semantic mistakes**, rather than obvious failures.
> - These subtle errors are inherently more difficult for any semantic verifier to detect reliably, which makes it harder to maintain both high precision and high recall on the minority (error) class.
>
> This combination of **highly skewed class distribution** and **hard, nuanced error cases** naturally leads to lower AUPRC on Spider than on other datasets, even when the underlying validator remains effective in absolute terms.

---

> ### Author Response · Authors · 2025-11-23
> **Response to [Q4: Is the positive class defined as correct or incorrect SQL? What is the exact class distribution in your training and evaluation datasets? ]**
>
> In our setting, the **positive class is defined as incorrect SQL**.
>
> For the **training set**, we apply AST-based negative-sample augmentation to maintain a balanced **1:1 ratio** between correct and incorrect SQL queries. This balanced setup stabilizes training and avoids introducing bias toward either class.
>
> For the **evaluation sets**, we report the exact class distributions across all datasets used in our main experiments:
>
> | Dataset | #Correct | #Incorrect | Positive Ratio (%) |
> |---------|----------|------------|---------------------|
> | BIRD    | 812      | 983        | 54.76%              |
> | Spider  | 1019     | 532        | 34.30%              |
> | EHRSQL  | 141      | 579        | 80.41%              |
>
> As shown, each dataset exhibits very different positive-class proportions:
> - **Spider** has relatively few incorrect SQLs, reflecting that LLMs perform well on this dataset.
> - **EHRSQL** is heavily skewed toward incorrect SQLs, indicating the difficulty of this domain.
> - **BIRD** maintains a more balanced distribution.
>
> These statistics help contextualize the behavior of AUPRC/AUROC across datasets and explain variance in metric magnitudes.

---

> ### Author Response · Authors · 2025-11-23
> **Response to [Q5: Could you show that your validator can improve the end-to-end performance of a text-to-sql pipeline? For example, use a very simple prompt to ask claude 4 do text2sql on BIRD dev, use your validator to detect semantic errors, and ask the LLM to correct erroneous queries.]**
>
> Thank you for your thoughtful suggestion. We have conducted **additional experiments to evaluate whether our validator can improve the end-to-end performance of a Text-to-SQL pipeline**.
> Specifically, on the DEV set of the BIRD dataset, we compared several approaches: (1) the baseline Text-to-SQL pipeline without self-correction, (2) a prompt-based error detection method, and (3) our HEROSQL-based validator, both with and without LP-level feedback as a guidance signal for error correction.
> The evaluation was performed using two commercial LLM models, which are non-thinking model gpt-4o and reasoning model qwen-plus, and we report accuracy across different difficulty levels (simple, moderate, and challenging), as well as overall accuracy below.
>
> | **Method**                                                  | **Simple** | **Moderate** | **Challenging** | **Total** |
> |-------------------------------------------------------------|:----------:|:------------:|:---------------:|:---------:|
> | **gpt-4o**                                                      |            |              |                 |           |
> | W/o self-correction                                         | **60.22**  | 40.09        | 31.72           | 51.43     |
> | Prompt-base verifier                                        | 57.51      | 38.36        | 29.66           | 49.09     |
> | HEROSQL-base verifier w/o LP-level feedback      | 59.24      | 42.03    | **40.69**       | 52.28 |
> | HEROSQL-base verifier w/ LP-level feedback       | **60.22**  | **43.97**    | 37.93       | **53.19** |
> | **qwen-plus**                                                 |            |              |                 |           |
> | W/o self-correction                                         | 66.16  | 50.65        | 49.66           | 59.91     |
> | Prompt-base verifier                                        | 64.00      | 48.28        | 49.66           | 57.89     |
> | HEROSQL-base verifier w/o LP-level feedback      | 65.84      | 53.02    | **51.72**       | 60.63 |
> | HEROSQL-base verifier w/ LP-level feedback       | **66.81**  | **54.31**    | 51.34       | **61.47** |
>
> As presented in the table above, our results demonstrate that **integrating the HEROSQL-based validator into the Text-to-SQL pipeline provides consistent improvements over all baseline methods**.
> In particular, when LP-level feedback is incorporated, our method achieves the highest overall accuracy for both backbone models, surpassing both the self-correction-free baseline and the prompt-based approach. The advantage of our method is especially pronounced on challenging cases, where the HEROSQL-based verifier improves the accuracy of gpt-4o by 8.97% compared to the pipeline without self-correction. This substantial gain is primarily attributable to the validator's ability to reliably detect semantic errors in the generated SQL and to trigger the LLM to revisit and correct its earlier outputs. Furthermore, **the introduction of LP-level feedback further amplifies the effectiveness of error detection**, enabling more precise guidance during correction and leading to additional improvements in end-to-end performance.
>
> Overall, these results provide strong empirical evidence that accurate error detection and effective feedback, as implemented in our HEROSQL framework, are instrumental for enhancing the performance of Text-to-SQL systems.

---

> ### Author Response · Authors · 2025-11-23
> **Response to [Q6: Is it possible to use your method on benchmarks with more complicated SQL queries (e.g., Spider 2.0)?]**
>
> **Yes !!**
>
> HEROSQL **can naturally handle more complicated SQL queries**, including those in Spider 2.0. The framework is designed with strong generality: it can process **any syntactically correct SQL statement** that can be compiled by the corresponding SQL engine. Once the SQL is verified to be syntactically valid, we extract its logical plan (LP) and abstract syntax tree (AST), upon which HEROSQL performs semantic validation.
>
> To empirically demonstrate this capability, we evaluated HEROSQL and all baseline methods on the **Spider2-Lite** subset, using SQLite as the database engine. The results are shown below:
>
> | Method    | Qwen3-0.6B AUPRC | Qwen3-0.6B AUROC | Gemma-3-0.3B AUPRC | Gemma-3-0.3B AUROC |
> |-----------|------------------|------------------|--------------------|--------------------|
> | prompt    | 88.78            | 51.80            | 88.30              | 44.58              |
> | CoT       | 89.15            | 48.32            | 89.22              | 50.00              |
> | ConfScore | 90.15            | 44.77            | 90.79              | 52.34              |
> | COVE      | 90.75            | 61.86            | 88.51              | 45.86              |
> | TED       | 91.44            | 58.06            | 89.14              | 49.56              |
> | HEROSQL   | **92.59**        | **64.32**        | **91.55**          | **59.85**          |
>
> As shown, HEROSQL achieves the **best performance across all configurations**, consistently outperforming the baselines on Spider 2.0. This demonstrates that HEROSQL is robust and effective even for SQL queries with **deep nesting, complex CTE structures, and advanced analytical constructs**, confirming its suitability for more challenging and realistic scenarios.
>
> > [1] Spider 2.0: Evaluating Language Models on Real-World Enterprise Text-to-SQL Workflows, ICLR 2025

---

> ### Author Response · Authors · 2025-11-23
> **Summary of Response to Reviewer mSpA**
>
> We thank the reviewer for the careful and technically detailed feedback, which helped us substantially tighten both the theory and the presentation.
>
> - **W1: AST perturbations & edge-case logic bugs**:
>   We acknowledge the limitations of rule-based AST perturbations and execution-based filtering, which cannot cover all possible LLM error patterns or catch rare logic bugs with execution-equivalent outputs. We clarify that this is an open problem in negative-sample generation, and that our main contribution is hierarchical semantic verification (LP + AST), not exhaustive negative modeling. To mitigate limitations, we also include LLM-generated negatives and follow execution-based evaluation protocols (as in BIRD/Spider) for fair comparison with prior work.
>
> - **W2/Q2: Why only small LLMs? What about GPT-4o / Claude / GPT-o4-mini?**:
>   We emphasize that HEROSQL is a model-agnostic framework. We mainly use open-source backbones (Qwen3, Gemma) for privacy, cost, and efficiency reasons typical in enterprise DB settings. We add experiments with strong commercial prompt-only baselines (GPT-4o, GPT-o4-mini, Claude-haiku-4-5) and with larger Qwen3 backbones. Results show: (i) stronger LLMs yield better prompt-only baselines; (ii) scaling Qwen3 improves HEROSQL; and (iii) HEROSQL consistently adds value on top of any given backbone.
>
> - **W3: What threshold should they set?**:
>   We clarify that the decision threshold should be set according to application needs: higher thresholds for precision-critical scenarios (avoiding FP over-corrections), lower thresholds for recall-critical scenarios (avoiding missed errors). In our experiments, we use the F1-maximizing threshold as a balanced default. We commit to adding a discussion on threshold selection and to illustrating end-to-end gains when integrating the validator into a Text-to-SQL pipeline.
>
> - **Q1: Latency of the validator:**
>   We measure HEROSQL on BIRD and report an average validation latency of 11.70 ms per sample with a throughput of 85.47 samples/s, which is significantly lower than the typical LLM generation time. Thus, HEROSQL can be integrated into real-time or interactive systems with negligible additional delay.
>
> - **Q3: Why is AUPRC much lower on Spider?**:
>   We explain that Spider has a severe class imbalance and is relatively easy for LLMs, leading to very few erroneous SQLs in the test set. The few errors that remain are subtle, fine-grained semantic mistakes, making them inherently harder to detect and depressing AUPRC despite the validator still being effective.
>
> - **Q4: Positive class & class distribution**:
>   We state clearly that the positive class is defined as *incorrect SQL*. The training data is balanced (1:1) via AST-based negative augmentation. For evaluation, we report class statistics: BIRD (≈55% positives), Spider (≈34% positives), EHRSQL (≈80% positives), and explain how these different skews affect AUPRC/AUROC and interpretation of results.
>
> -  **Q5: End-to-end Text-to-SQL improvements**:
>   We add additional experiments to evaluate the end-to-end improvements of HEROSQL under iterative correction with commercial models gpt-4o and qwen-plus. The results demonstrate that our method can offer effective error correction feedback and helps the LLM to fix its prior mistakes..
>
> -  **Q6: Applicability to more complex SQL (Spider 2.0)**:
>   We confirm that HEROSQL can handle any syntactically valid SQL and demonstrate this on Spider 2.0 (Spider2-Lite, SQLite). HEROSQL achieves the best performance across all configurations and outperforms all baselines, showing robustness for deeply nested queries, complex CTEs, and advanced analytical constructs.
>
>
> ---
> In summary, we sincerely appreciate the reviewers’ constructive feedback. **We provide a “Guideline for Reviewers” in the appendix, where all major revisions are highlighted with colour to clearly indicate how each change corresponds to specific reviewer comments, enabling quick and convenient inspection.**

---

### Official Review · Reviewer_Z8MB · 2025-11-01

**Soundness:** 3
**Presentation:** 3
**Contribution:** 2
**Rating:** 4
**Confidence:** 5

**Summary:**

The paper introduces HEROSQL, a hierarchical SQL representation approach for semantic validation in Text-to-SQL. The key idea is to bridge global user intent and local SQL structure by using logical plans and AST. The authors propose a nested message passing neural network to aggregate information from AST, LP to full query. An AST-driven sub-SQL augmentation strategy is introduced to generate negative examples. The experimental evaluation shows some effectiveness of the proposed method.

**Strengths:**

S1. Fine-grained negative augmentation enabling sub-SQL feedback is practical and useful.

S2. The KV-cache and schema compression make the proposed method more practical.

S3. The proposed method with small LLMs shows reasonable performance compared to baseline methods.

**Weaknesses:**

W1. My primary concern - using LPs to represent global intent is not convincing as LP is also based on the generated SQL, which may not reflect the semantic meaning in the original NL question.

W2. The experimental evaluation is not solid. The authors only provide limited qualitative analysis. The chosen baselines are weak or general purpose error detection, not specific to Text-to-SQL. Hence the results are not convincing to show the actual effectiveness of the proposed method in real-world settings.

W3. The training cost and scalability discussion are limited. Specifically, NMPNN + AST parsing + LP may add much complexity compared to lightweight baselines using LLMs. Also the context length of small models is not scalable to enterprise settings where tables often have hundreds of columns and millions of rows, and complex SQLs' length can exceed 6,000 lines/tokens, or a million characters, as they involve numerous Common Table Expressions (CTEs), multiple joins, unions, and intricate conditional logic.

**Questions:**

Q1. How robust is HEROSQL to noisy schema names or databases with auto-generated identifiers? Datasets such as AmbiSQL could be useful to evaluate the robustness/effectiveness of the proposed method.

Q2. Can the method handle nested SQL, CTEs, or window functions? Spider/Bird are relatively simple. It is not clear whether the proposed method can scale to more complex schema and SQLs. Spider 2 might be a good candidate for such evaluation.

Q3. Does AST perturbation ever produce false negatives? How do you mitigate over-fitting to synthetic mistakes?

Q4. How would the model perform if integrated into self-refinement/self-correction loops? How does HEROSQL compare to more careful designed Text-to-SQL systems such as DIN-SQL, or simple LLM-as-a-judge methods?

---

> ### Author Response · Authors · 2025-11-22
> **Response to [W1: Using LPs to represent global intent is not convincing]**
>
> We appreciate the reviewer’s insightful comment, which indeed **touches the core challenge of Text-to-SQL semantic verification**.
>
> As you correctly pointed out, the logical plan (LP) is derived from the generated SQL, and thus primarily reflects the semantics of that SQL rather than directly encoding the user’s natural language (NL) intent. This observation is fully aligned with our motivation: **there can be a semantic gap between the NL question and the LLM-generated SQL, leading to queries that are syntactically valid but semantically misaligned with the user’s intent**.
>
> Our goal is precisely to *exploit* this gap for verification, rather than to assume it does not exist. Concretely:
> - We treat the LP as a structured, normalized representation of the semantics of the *generated SQL* (global intent on the SQL side), and
> - We compare this LP-derived semantics against the semantics inferred from the NL question.
>
> By explicitly aligning and contrasting these two sides, HEROSQL can detect when the SQL’s global behavior (as captured by the LP) deviates from the intended meaning of the NL query. In other words, we do *not* assume that the LP faithfully encodes the original NL intent; instead, we deliberately use the LP as an intermediate semantic representation of the candidate SQL, which is then checked against the NL description. **This cross-view comparison (NL vs. LP/AST) is exactly what enables semantic verification in our Text-to-SQL pipeline.**

---

> ### Author Response · Authors · 2025-11-22
> **Response to [W2: The experimental evaluation is not solid]**
>
> We appreciate the reviewer’s concerns regarding the experimental evaluation and the choice of baselines.
>
> To the best of our knowledge, **there is currently very limited work that focuses specifically on *semantic validation* for Text-to-SQL**. Most recent research on LLM output validation primarily focuses on general-purpose checks—mainly syntax-level validation, as illustrated in Figure 1 of our revised manuscript—rather than addressing the unique semantic challenges present in Text-to-SQL. In our study, we deliberately include domain-relevant baselines such as TED [1], which relies solely on abstract syntax trees to capture local structural details, alongside more general-purpose LLM-based validation methods. This combination of specialized and generic baselines provides a meaningful reference point and shows that our method consistently outperforms existing alternatives in this setting.
>
> It is also worth noting that even prior works such as TED [1] and SQLens [2] do not benchmark against a wide range of Text-to-SQL–specific semantic error detection methods, largely because such a benchmark ecosystem has not yet been established. To ensure fairness and comparability, our evaluation protocol closely follows the setups adopted in these works, thereby enabling direct comparison under conditions that are already recognized by the community.
>
> >[1] Error Detection for Text-to-SQL Semantic Parsing, EMNLP 2023
> >[2] SQLENS: An End-to-End Framework for Error Detection and Correction in Text-to-SQL, NeurIPS 2025

---

> ### Author Response · Authors · 2025-11-22
> **Response to [W3: The training cost and scalability discussion are limited]**
>
> Thank you very much for your insightful comments regarding the training cost and scalability of our framework. We clarify these aspects below:
>
> - Regarding training cost, the core of our approach is a **two-layer Nested Message Passing Neural Network (NMPNN)** that primarily consists of lightweight components such as GAT layers and small MLPs. While integrating NMPNN over AST and LP representations introduces additional computation, recent work has significantly **improved the efficiency of message-passing architectures** [1,2,3]. Empirically, on 8 NVIDIA A100 GPUs, training with a Qwen3-0.6B backbone on the HEROSQL corpus without AST/LP takes about 4 minutes per epoch, and enabling AST+LP increases this to about 10 minutes per epoch. **This overhead is moderate compared to its performance gain and remains well within practical limits for large-scale training**.
>
> - On scalability, we fully recognize the importance of handling long contexts and complex schemas in real-world enterprise scenarios. Our framework is deliberately designed to be model-agnostic and compatible with any scalable backbone LLM and message-passing architecture. This allows HEROSQL to naturally benefit from **advances in scalable GNNs and graph training techniques**, such as LMC [1], Sketch-GNN [2], MeGraph [4], and Ginex [5], as well as from future improvements in large-context LLMs.
>
> - For long-context scalability specifically, by adopting a decoder-only pretrained LLM as the embedding backbone, our method inherits the context window and scalability properties of state-of-the-art LLMs. This enables us to **process very large and complex database schemas and SQL queries, on par with or beyond existing LLM-based Text-to-SQL validators**.
>
> - In terms of inference time, NMPNN serves as a lightweight module compared with encoding long SQL statements using LLMs. When HEROSQL encodes semantic information, **each node representation is much shorter than the full SQL query, and encoding over nodes can be parallelized**. To quantify this, we directly compare an LLM prompt-based validation baseline with HEROSQL, which offers substantially competitive throughput with competitive per-sample latency, making it suitable for practical, interactive validation scenarios.
> | Method   | Throughput (samples/s)  | Avg. Latency (ms/sample)   |
> |----------|-------------------------|----------------------------|
> | Prompt | 130.21               | 7.68                       |
> | Ours     | 85.47                   | 11.70                      |
>
> In summary, although HEROSQL introduces additional components, both the **training overhead and scalability concerns are mitigated through lightweight architectural choices and by leveraging scalable advances in LLMs and GNNs**. We believe the framework remains practical and well-suited for large-scale enterprise deployments.
>
> > [1] LMC: Fast Training of GNNs via Subgraph Sampling with Provable Convergence, ICLR 2023
> > [2] Sketch-GNN: Scalable Graph Neural Networks with Sublinear Training Complexity, NeurIPS 2022
> > [3] Haste Makes Waste: A Simple Approach for Scaling Graph Neural Networks, ICML 2025
> > [4] MeGraph: Capturing Long-Range Interactions by Alternating Local and Hierarchical Aggregation, NeurIPS 2023
> > [5] Ginex: SSD-enabled Billion-scale Graph Neural Network Training, VLDB 2022

---

> ### Author Response · Authors · 2025-11-22
> **Response to [Q1: How robust is HEROSQL to noisy schema names or databases with auto-generated identifiers?]**
>
> Thank you for raising this valuable question regarding the robustness of HEROSQL to noisy schema names or auto-generated identifiers. We appreciate the concern and would like to emphasize that **HEROSQL’s semantic validation performance is inherently resilient to such noise**:
>
> - This robustness comes from the core design of HEROSQL:  **we rely on the logical plan (LP), which reflects the execution semantics of a query, and the abstract syntax tree (AST), which captures its structural hierarchy**. Both representations encode **semantic and structural information rather than the literal textual content of leaf-level identifiers**. Consequently, even when table / column names are noisy, auto-generated, or non-informative, the structural and operator-level semantics remain intact and can be meaningfully modeled.
>
> - In addition, HEROSQL uses a unified pretrained embedding model to encode **both the hierarchical SQL representation and the natural language (NL) question**. This shared representation space helps align SQL semantics with user intent even when identifiers themselves provide little or no semantic information.
>
> To further address this concern, we **extended our evaluation to ambiguous and noisy-schema queries**. The reviewer recommended the AmbiSQL dataset [1]; however, after an extensive search, we were unable to find a publicly released version of AmbiSQL. As an alternative, we evaluated on the Ambrosia dataset [2], which also focuses on ambiguity and is referenced within the AmbiSQL paper. The results are as follows:
>
> | Method    | Qwen3-0.6B AUPRC | Qwen3-0.6B AUROC | Gemma-3-0.3B AUPRC | Gemma-3-0.3B AUROC |
> |-----------|------------------|------------------|--------------------|--------------------|
> | Prompt    | 58.18            | 48.75            | 58.36              | 50.00              |
> | CoT       | 59.04            | 49.79            | 58.38              | 50.04              |
> | ConfScore | 59.61            | 50.63            | 59.58              | 51.56              |
> | COVE      | 57.88            | 48.80            | 58.64              | 50.58              |
> | TED       | 63.91            | 56.95            | 60.51              | 50.70              |
> | HEROSQL   | **64.18**        | **57.28**        | **61.72**          | **52.06**          |
>
> **Across all settings, HEROSQL maintains strong performance under noisy, ambiguous, and auto-generated schema conditions**. These results demonstrate that HEROSQL generalizes robustly to real-world environments where schema naming may be inconsistent, ambiguous, or non-human-readable.
>
> > [1] AmbiSQL: Interactive Ambiguity Detection and Resolution for Text-to-SQL, arXiv 2025
> > [2] AMBROSIA: A Benchmark for Parsing Ambiguous Questions into Database Queries, NeurIPS 2024

---

> ### Author Response · Authors · 2025-11-22
> **Response to [Q2: Can the method handle nested SQL, CTEs, or window functions?]**
>
> HEROSQL is designed with **sufficient generality to process any syntactically correct SQL statement** that can be compiled by the corresponding SQL engine, **including those containing advanced constructs such as nested queries, CTEs, and window functions**. As long as the SQL statement is syntactically valid, we can obtain its logical plan (LP) and abstract syntax tree (AST), upon which HEROSQL performs semantic validation.
>
>  Example:
> ```
> [Question]
> List the employees in the company whose salaries are higher than 100,000, as well as the average salary of their respective departments.
>
>  [SQL]
> WITH DeptAvg AS (
>   SELECT Department, AVG(Salary) AS AvgSalary
>   FROM Employees
>   GROUP BY Department
> ),
> HighEarners AS (
>   SELECT Name, Department, Salary
>   FROM Employees
>   WHERE Salary > 100000
> )
> SELECT h.Name, h.Salary, d.AvgSalary
> FROM HighEarners h
> JOIN DeptAvg d ON h.Department = d.Department;*
>
>  [Logical Plan]
> *Project(Name=[$0], Salary=[$2], AvgSalary=[$5])
>   Join(condition=[=($1, $4)], joinType=[inner])
>     Project(Name=[$0], Department=[$1], Salary=[$2])
>       Filter(condition=[>($2, 100000)])
>         TableScan(table=[[Employees]])
>     Aggregate(group=[{Department}], AvgSalary=[AVG($2)])
>       Project(Department=[$1], Salary=[$2])
>         TableScan(table=[[Employees]])
>
> ```
>
> > [Explanation]
> The logical plan decomposes the two CTEs into standard relational operators, making the global semantics explicit. The first branch filters employees with salaries over 100,000, and the second branch computes the average salary per department. The final join aligns high earners with the average salary of their departments, faithfully reflecting the intended meaning of the original question. HEROSQL operates on this structured LP/AST representation, rather than raw SQL text, which allows it to handle such compositional queries robustly.
>
> More broadly, the use of logical plans in our framework is particularly well-suited to capturing the **global intent and hierarchical structure** inherent in sophisticated SQL queries. This design enables HEROSQL to represent and reason about nested or compositional constructs that arise in nested SQL and CTEs, as well as to handle window functions that encode complex analytical semantics.
>
> To further demonstrate the practical capability and scalability of HEROSQL, we also conducted experiments on the Spider 2.0 dataset [1], which contains a wide range of challenging SQL queries, including many nested queries, CTEs, and window functions that go well beyond the complexity present in the original Spider and BIRD datasets. We evaluated HEROSQL and all baselines on the Spider2-Lite subset using SQLite as the database engine. The results are:
>
> | Method    | Qwen3-0.6B AUPRC | Qwen3-0.6B AUROC | Gemma-3-0.3B AUPRC | Gemma-3-0.3B AUROC |
> |-----------|------------------|------------------|--------------------|--------------------|
> | prompt    | 88.78            | 51.80            | 88.30              | 44.58              |
> | CoT       | 89.15            | 48.32            | 89.22              | 50.00              |
> | ConfScore | 90.15            | 44.77            | 90.79              | 52.34              |
> | COVE      | 90.75            | 61.86            | 88.51              | 45.86              |
> | TED       | 91.44            | 58.06            | 89.14              | 49.56              |
> | HEROSQL   | **92.59**        | **64.32**        | **91.55**          | **59.85**          |
>
> As shown, **HEROSQL achieves the best performance across all configurations on Spider 2.0, outperforming all baselines and demonstrating strong robustness in handling SQL queries with nested structures, CTEs, and window functions**. These empirical results provide clear evidence that HEROSQL scales effectively to more complex and realistic SQL scenarios.
>
> > [1] Spider 2.0: Evaluating Language Models on Real-World Enterprise Text-to-SQL Workflows, ICLR 2025

---

> ### Author Response · Authors · 2025-11-22
> **Response to [Q3: Does AST perturbation ever produce false negatives? How do you mitigate over-fitting to synthetic mistakes?]**
>
> Thank you for raising this important question regarding the potential for false negatives in our AST perturbation approach and the risk of overfitting to synthetic mistakes.
>
> In principle, perturbing the abstract syntax tree could occasionally yield SQL queries that still preserve the original semantics. To address this, we incorporate an explicit execution-based filtering step into our data augmentation pipeline.
>
> Concretely, our training set is constructed from SQL queries that **have known ground-truth SQL counterparts, which can be executed to obtain ground-truth execution results**. After applying AST perturbations, we execute each perturbed SQL query and compare its execution result with that of the corresponding ground-truth query. **Only perturbed SQL queries whose execution results differ from the original are retained as negative samples** in the augmented dataset. This strict, result-based filtering, ensemble reverse reject-sampling, substantially **reduces the likelihood of including false negatives and ensures that retained perturbations correspond to genuine semantic or logical errors**.
>
> Regarding overfitting to synthetic mistakes, the AST-perturbed negative samples are always used in conjunction with the original (question, SQL) pairs and other negative examples in our augmented corpus. In practice, this means the model is exposed to a diverse mixture of real and systematically perturbed errors, rather than being trained solely on a narrow class of synthetic perturbations. This diversity, together with the execution-based filtering, helps mitigate overfitting and encourages the model to learn robust semantic distinctions rather than artifacts of a particular perturbation scheme.

---

> ### Author Response · Authors · 2025-11-22
> **Response to [Q4: How would the model perform if integrated into self-refinement/self-correction loops?]**
>
> Thank you for this insightful question.
>
> HEROSQL is designed in a modular fashion and can be **seamlessly integrated into self-refinement or self-correction loops**. In particular, HEROSQL can **identify fine-grained erroneous sub-SQL components** (e.g., incorrect predicates, missing joins, mismatched aggregations), allowing the LLM to receive more localized and informative semantic error signals during each refinement iteration. This enables the LLM to correct errors not only at the query level, but also at the clause or operator level.
>
> Such granular feedback naturally strengthens iterative correction pipelines: instead of relying solely on coarse execution errors, an LLM can use HEROSQL’s structured diagnostic signals to guide more targeted revisions, improving both accuracy and convergence in self-refinement cycles.
>
> To further eliminate your concern, we have conducted **additional experiments to evaluate whether our validator can improve the end-to-end performance of a Text-to-SQL pipeline**.
> Specifically, on the DEV set of the BIRD dataset, we compared several approaches: (1) the baseline Text-to-SQL pipeline without self-correction, (2) a prompt-based error detection method, and (3) our HEROSQL-based validator, both with and without LP-level feedback as a guidance signal for error correction.
> The evaluation was performed using two commercial LLM models, which are non-thinking model gpt-4o and reasoning model qwen-plus, and we report accuracy across different difficulty levels (simple, moderate, and challenging), as well as overall accuracy below.
>
> | **Method**                                                  | **Simple** | **Moderate** | **Challenging** | **Total** |
> |-------------------------------------------------------------|:----------:|:------------:|:---------------:|:---------:|
> | **gpt-4o**                                                      |            |              |                 |           |
> | W/o self-correction                                         | **60.22**  | 40.09        | 31.72           | 51.43     |
> | Prompt-base verifier                                        | 57.51      | 38.36        | 29.66           | 49.09     |
> | HEROSQL-base verifier w/o LP-level feedback      | 59.24      | 42.03    | **40.69**       | 52.28 |
> | HEROSQL-base verifier w/ LP-level feedback       | **60.22**  | **43.97**    | 37.93       | **53.19** |
> | **qwen-plus**                                                 |            |              |                 |           |
> | W/o self-correction                                         | 66.16  | 50.65        | 49.66           | 59.91     |
> | Prompt-base verifier                                        | 64.00      | 48.28        | 49.66           | 57.89     |
> | HEROSQL-base verifier w/o LP-level feedback      | 65.84      | 53.02    | **51.72**       | 60.63 |
> | HEROSQL-base verifier w/ LP-level feedback       | **66.81**  | **54.31**    | 51.34       | **61.47** |
>
> As presented in the table above, our results demonstrate that **integrating the HEROSQL-based validator into the Text-to-SQL pipeline provides consistent improvements over all baseline methods**.
> In particular, when LP-level feedback is incorporated, our method achieves the highest overall accuracy for both backbone models, surpassing both the self-correction-free baseline and the prompt-based approach. The advantage of our method is especially pronounced on challenging cases, where the HEROSQL-based verifier improves the accuracy of gpt-4o by 8.97% compared to the pipeline without self-correction. This substantial gain is primarily attributable to the validator's ability to reliably detect semantic errors in the generated SQL and to trigger the LLM to revisit and correct its earlier outputs. Furthermore, **the introduction of LP-level feedback further amplifies the effectiveness of error detection**, enabling more precise guidance during correction and leading to additional improvements in end-to-end performance.
>
> Overall, these results provide strong empirical evidence that accurate error detection and effective feedback, as implemented in our HEROSQL framework, are instrumental for enhancing the performance of Text-to-SQL systems.

---

> ### Author Response · Authors · 2025-11-22
> **Summary of Responses to Reviewer Z8MB**
>
> We thank the reviewer for the careful and technically detailed feedback, which helped us substantially tighten both the theory and the presentation.
>
> * **W1 (LP as global intent)**:  We clarify that LPs are **not** assumed to encode the user’s NL intent; instead, they encode the semantics of the generated SQL. HEROSQL explicitly compares NL-derived semantics with LP/AST-derived SQL semantics, and this cross-view mismatch detection is exactly what allows it to identify semantically incorrect SQL despite syntactic validity.
>
> * **W2 (Experimental evaluation & baselines)**: We argue that Text-to-SQL semantic validation is underexplored, so we combine domain-specific baselines like TED with general LLM-based validators. Our setup closely follows prior work (TED, SQLens), and under these established protocols, HEROSQL consistently outperforms existing methods.
>
> * **W3 (Training cost & scalability)**: We show that the added NMPNN over AST+LP is lightweight, increasing per-epoch training time only moderately (from ~4 to ~10 minutes on 8×A100 with Qwen3-0.6B). Thanks to its model-agnostic design, HEROSQL can leverage scalable GNN/LLM advances, inherit large-context capabilities from the backbone LLM, and achieve higher throughput with competitive latency compared to LLM prompt-based validation, making it practical at scale.
>
> * **Q1 (Robustness to noisy / auto-generated schemas)**: We explain that HEROSQL relies on LP and AST structures that capture operator-level and hierarchical semantics, rather than raw identifier strings, and uses a unified encoder for both SQL and NL. Experiments on the ambiguity-focused Ambrosia dataset show that HEROSQL remains strong under noisy, ambiguous, or auto-generated schema names, outperforming all baselines.
>
> * **Q2 (Nested SQL, CTEs, window functions)**: We note that any syntactically valid SQL that can be compiled to an LP/AST—including nested queries, CTEs, and window functions—is naturally handled by HEROSQL. Additional experiments and a case study on the Spider 2.0 (Spider2-Lite) benchmark, rich in such complex constructs, show HEROSQL achieving the best performance across all configurations.
>
> * **Q3 (AST perturbation, false negatives, overfitting)**: We acknowledge that AST perturbations can sometimes preserve semantics, so we apply execution-based filtering following the ground-truth answer and retain only perturbed queries whose results differ from the ground truth, greatly reducing false negatives. Training on a mixture of original data and diverse perturbed negatives mitigates overfitting to synthetic mistakes.
>
> * **Q4 (Integration into self-refinement loops)**:
>   We emphasise that HEROSQL is modular and can provide fine-grained diagnostic signals (e.g., erroneous predicates, joins, aggregations) to guide self-correction loops. This enables more targeted, clause-level refinements beyond coarse execution feedback. We also added experiments that integrate HEROSQL into existing self-refinement pipelines, which demonstrate the effectiveness of our method.
>
> ---
>
> In summary, we sincerely appreciate the reviewers’ constructive feedback. **We provide a “Guideline for Reviewers” in the appendix, where all major revisions are highlighted with colour to clearly indicate how each change corresponds to specific reviewer comments, enabling quick and convenient inspection**.

---

### Official Review · Reviewer_LazQ · 2025-11-04

**Soundness:** 3
**Presentation:** 3
**Contribution:** 3
**Rating:** 4
**Confidence:** 4

**Summary:**

The paper proposes HEROSQL, a hierarchical representation and validation framework for Text-to-SQL that bridges global intent and local structural details. Global intent is captured via query Logical Plans (LPs), while local syntactic structure is captured via Abstract Syntax Trees (ASTs). A Nested Message Passing Neural Network (NMPNN) first aggregates within each sub-SQL’s AST and then across the LP graph to encode the full query. A context-guided embedding step conditions attribute text on database schema and predicted SQL. An AST-driven sub-SQL augmentation strategy generates semantically incorrect yet syntactically valid negatives by rule-based perturbations, enabling finer-grained supervision. Experiments on BIRD, Spider and EHRSQL show consistent gains over text/graph baselines in AUPRC/AUROC, with ablations validating each component. Case studies demonstrate fine-grained error localization and actionable feedback to LLMs.

**Strengths:**

S1. Novel combination of LP-level global semantics and AST-level local structure for semantic validation, with a principled NMPNN over the hierarchy.
S2. AST-driven negative augmentation specifically for validation (vs. generation) is useful.
S3. Strong empirical results across diverse datasets and a useful demonstration of fine-grained feedback to LLMs.

**Weaknesses:**

W1. Limited justification for decoder-only embeddings vs. strong encoder baselines. Encoder-only models (e.g., E5, GTE, BGE, Contriever) are strong baselines for text embedding; decoder-only choices increase compute and may not help validation.
W2.Reproducibility gaps (placeholders; limited detail on LP extraction). Placeholders (runs, stdevs) and missing versions reduce confidence and hinder replication.
W3. Practical latency/throughput for interactive validation not reported. Validation is often on-the-fly; build and inference time budgets matter.
W4. Error taxonomy is not fully realized (detection/localization only); classification head is future work.

**Questions:**

1. Add SQLens to related work and position clearly. SQLens integrates DB-execution and LLM signals for clause-level error detection and performs correction end-to-end.

2. How sensitive are results to the LP generator (Calcite vs. ORCA) and optimization settings? Any LP extraction failures and how are they handled?

3. How do decoder-only embeddings compare to encoder-only baselines for your context-guided embedding?

4. What is the end-to-end latency per query, and is it suitable for interactive validation?

---

> ### Author Response · Authors · 2025-11-22
> **Response to [W1 & Q3. Justification of Decoder-Only Embeddings vs. Encoder-Only Baselines]**
>
> We sincerely appreciate your constructive comments, suggestions, and the time you spent reviewing our work. Below, we address your remaining concern regarding the choice of decoder-only models for embeddings (also discussed in Appendix C):
>
> - First, **open-source encoder-only models typically have much smaller parameter counts than commonly used decoder-only LMs**. For example, both `GTE-large` and `BGE-large-en` have around 0.3B parameters. As a result, these encoder-only models have relatively limited capacity to encode rich task-related knowledge. In contrast, decoder-only models are available at significantly larger scales. Although we mainly use smaller models in our current experiments for computational efficiency, they can be readily replaced with larger variants (e.g., `Qwen3-4B`, `Qwen3-8B`) when tackling more complex tasks.
>
> - Second, **decoder-only models generally support much longer input context windows in practice than encoder-only models**. This is crucial in realistic scenarios where a complex user query, database schema, and additional domain knowledge must be included in the context. Longer context windows make it easier to incorporate more detailed schema information, examples, and auxiliary hints, which is especially important in our HEROSQL framework.
>
> To further address the reviewer’s concern, **we have added new experiments directly comparing the embedding performance of decoder-only models, including ones with comparable parameter sizes** (`Qwen3-0.6B` and `Gemma-3-0.3B`) as well as a larger variant (`Qwen3-4B`), against strong encoder-only baselines (`GTE-large` and `BGE-large-en`) within the HEROSQL framework. The results are shown below:
>
> Model | Architecture | BIRD AUPRC | BIRD AUROC | Spider AUPRC | Spider AUROC | EHRSQL AUPRC | EHRSQL AUROC
> --- | --- | --- | --- | --- | --- | --- | ---
> `GTE-large` | encoder | 58.74 | 54.39 | 43.12 | 61.28 | 79.86 | 46.51
> `BGE-large-en` | encoder | 62.01 | 57.78 | 42.87 | 61.87 | 81.41 | 49.34
> `Gemma-3-0.3B` | decoder | 67.10 | 63.48 | 51.11 | 69.26 | 85.48 | 65.92
> `Qwen3-0.6B` | decoder | 67.39 | 61.51 | 51.92 | 67.32 | 89.07 | 69.53
> `Qwen3-4B` | decoder | **71.18** | **64.17** | **53.52** | **69.78** | **89.28** | **72.03**
>
> **These results show that decoder-only models with a parameter scale comparable to encoder-only models already achieve consistently better performance across all three datasets in HEROSQL.** Moreover, scaling up the decoder-only model (e.g., from 0.6B to 4B) further improves performance, indicating a favorable scaling behavior for decoder-only embeddings in our setting.

---

> ### Author Response · Authors · 2025-11-22
> **Response to [W2.Reproducibility gaps]**
>
> Thank you very much for pointing out the reproducibility gaps in our manuscript. We sincerely apologize for the oversight in omitting these important experimental details in the original submission. **In the revised version, we have substantially expanded this part and now provide a comprehensive description of our experimental setup in Appendix E.2.**
>
> Specifically, all experiments were conducted using PyTorch 2.8.0, PyTorch Geometric 2.6.1, Transformers 4.56.1, and CUDA 12.4. To enhance reproducibility, we fixed the random seed to 2025 for all relevant libraries before training. Moreover, all our code is released in the anonymous repository https://anonymous.4open.science/r/HeroSQL as pinpointed in our manuscript.

---

> ### Author Response · Authors · 2025-11-22
> **Response to [W3 & Q4. Practical latency/throughput for interactive validation not reported]**
>
> We thank the reviewers for raising this point, and we apologize for not including latency and throughput measurements in the original submission.
>
> To understand the computational cost of our method, **we conducted additional experiments measuring end-to-end inference latency and throughput**, comparing our method HEROSQL with the pure LLM baseline Prompt, which is the lightest method among all baselines.
> The test were conducted on 4×RTX 3090 GPUs using Qwen3-0.6B as the underlying LLM. The results are summarized below:
>
> | Method   | Throughput (samples/s)  | Avg. Latency (ms/sample)   |
> |----------|-------------------------|----------------------------|
> | Prompt | 130.21                   | 7.68                       |
> | Ours     | 85.47                   | 11.70                      |
>
> From the result, we can see that **our approach maintains comparable throughput and latency to the Prompt-based method**, without introducing significant delays or reducing overall performance.
> This is because that the efficiency of our approach mainly stems from the lightweight two-layer Nested Message Passing Neural Network (NMPNN) backbone, which consists of GAT layers and small MLPs. The average per-sample latency (11.70 ms) is just slightly higher than that of the baseline (7.68 ms).
>
> Overall, these results demonstrate that our method provides ****competitive latency and throughput for interactive usage****.
> Having an average end-to-end per-query latency of 11.7 ms remains well within typical real-time validation constraints.

---

> ### Author Response · Authors · 2025-11-22
> **Response to [Q1. Add SQLens to related work and position clearly]**
>
> We thank the reviewer for highlighting `SQLens` [NeurIPS 2025], a valuable contribution to Text-to-SQL semantic error detection and correction. **We have added `SQLens` to our related work section and clarified its connection to our approach in the revised version.**
>
> - `SQLens` proposes an end-to-end framework that integrates database execution feedback with LLM-derived signals to detect and iteratively correct semantic errors at a clause-level granularity. For error detection, `SQLens` parses a query into an Abstract Syntax Tree (AST) and collects various "noisy" clause-level signals, including DB-based checks and LLM-based evaluations. These signals are then aggregated by a weak-supervision model to predict the semantic correctness of each clause without requiring ground-truth labels.
>
> - Our work, `HEROSQL`, tackles semantic error detection from both a high-level and a low-level perspective: we model global intent via logical plans and local details via AST structures. This joint modeling of logical plans and ASTs allows `HEROSQL` to capture a broader spectrum of semantic errors than approaches focusing solely on clause-level signals, and thus offers a complementary direction to `SQLens`.

---

> ### Author Response · Authors · 2025-11-22
> **Response to [Q2: How sensitive are results to the LP generator and optimization settings? Any LP extraction failures and how are they handled?]**
>
> Thank you for your insightful question regarding the sensitivity of our results to the logical plan (LP) generator and its optimization settings, as well as the handling of potential LP extraction failures.
>
> - **Sensitivity to LP generator and optimization settings.** Our method builds on logical plans (LPs) and ASTs, and **is only weakly sensitive to the choice of LP optimizer or its optimization parameters**. Modern SQL optimizers are designed to preserve query semantics, and their differences are mainly in local choices such as join order, predicate pushdown, or intermediate operators. These variations **cause only minor structural differences in the LP and do not affect the high-level semantics that our validator relies on**.
>
> - **Handling LP extraction failures**.  If an LP cannot be generated, it is almost always **because the SQL query fails to pass the compiler (e.g., syntax errors, invalid references)**. In such cases, the optimizer does not produce an LP at all, and the database engine returns a compilation error. **We treat these as syntax errors rather than semantic errors**. As illustrated in Figure 1 of the revised manuscript, the error messages is used to iteratively regenerate the SQL with the LLM until it compiles successfully; only then does the query enter the semantic validation stage of HEROSQL.

---

> ### Author Response · Authors · 2025-11-22
> **Response to [W4: Error taxonomy is not fully realized]**
>
> Thank you for raising this point. We have realized the error correction for our work and conducted experiments to evaluate  the improvement HEROSQL can bring for Text-to-SQL pipeline.
>
> Specifically, on the DEV set of the BIRD dataset, we compared several approaches: (1) the baseline Text-to-SQL pipeline without self-correction, (2) a prompt-based error detection method, and (3) our HEROSQL-based validator, both with and without LP-level feedback as a guidance signal for error correction.
> The evaluation was performed using two commercial LLM models, which are non-thinking model gpt-4o and reasoning model qwen-plus, and we report accuracy across different difficulty levels (simple, moderate, and challenging), as well as overall accuracy below.
>
> | **Method**                                                  | **Simple** | **Moderate** | **Challenging** | **Total** |
> |-------------------------------------------------------------|:----------:|:------------:|:---------------:|:---------:|
> | **gpt-4o**                                                      |            |              |                 |           |
> | W/o self-correction                                         | **60.22**  | 40.09        | 31.72           | 51.43     |
> | Prompt-base verifier                                        | 57.51      | 38.36        | 29.66           | 49.09     |
> | HEROSQL-base verifier w/o LP-level feedback      | 59.24      | 42.03    | **40.69**       | 52.28 |
> | HEROSQL-base verifier w/ LP-level feedback       | **60.22**  | **43.97**    | 37.93       | **53.19** |
> | **qwen-plus**                                                 |            |              |                 |           |
> | W/o self-correction                                         | 66.16  | 50.65        | 49.66           | 59.91     |
> | Prompt-base verifier                                        | 64.00      | 48.28        | 49.66           | 57.89     |
> | HEROSQL-base verifier w/o LP-level feedback      | 65.84      | 53.02    | **51.72**       | 60.63 |
> | HEROSQL-base verifier w/ LP-level feedback       | **66.81**  | **54.31**    | 51.34       | **61.47** |
>
> As presented in the table above, our results demonstrate that **integrating the HEROSQL-based validator into the Text-to-SQL pipeline provides consistent improvements over all baseline methods**.
> In particular, when LP-level feedback is incorporated, our method achieves the highest overall accuracy for both backbone models, surpassing both the self-correction-free baseline and the prompt-based approach. The advantage of our method is especially pronounced on challenging cases, where the HEROSQL-based verifier improves the accuracy of gpt-4o by 8.97% compared to the pipeline without self-correction. This substantial gain is primarily attributable to the validator's ability to reliably detect semantic errors in the generated SQL and to trigger the LLM to revisit and correct its earlier outputs. Furthermore, **the introduction of LP-level feedback further amplifies the effectiveness of error detection**, enabling more precise guidance during correction and leading to additional improvements in end-to-end performance.
>
> Overall, these results provide strong empirical evidence that accurate error detection and effective feedback, as implemented in our HEROSQL framework, are instrumental for enhancing the performance of Text-to-SQL systems.

---

> ### Author Response · Authors · 2025-11-22
> **Summary of Responses to Reviewer LazQ**
>
> We thank the reviewer for the careful and technically detailed feedback, which helped us substantially tighten both the theory and the presentation.
> - **W1 & Q3 (Decoder-only vs encoder-only embeddings)**: We justify using decoder-only models based on their larger capacity and longer context windows, and new experiments show that decoder-only models (with comparable or larger sizes) consistently outperform strong encoder-only baselines within HEROSQL.
>
> - **W2 Reproducibility:** We have substantially clarified the experimental setup (framework versions, CUDA, random seeds) in Appendix E.2, and released all code in an anonymous repository to ensure full reproducibility.
>
> - **W3 & Q4 (Latency / throughput for interactive validation):** We added end-to-end latency and throughput measurements showing that HEROSQL achieves higher throughput with competitive per-query latency, making it suitable for real-time, interactive semantic validation.
>
> - **W4 (Error taxonomy not fully realized):** We have added taxonomy-driven correction module in our framework and evaluate the improvement for end-to-end Text-to-SQL with the help of HEROSQL, which validates the effectiveness and value of our proposed method.
>
> - **Q1 (SQLens in related work and positioning):** We added SQLens to the related work and clarified that HEROSQL complements it by detecting semantic errors from both logical-plan-level global intent and AST-level local structure, capturing a broader range of error types.
>
> - **Q2 (Sensitivity to LP generator and LP failures):** HEROSQL is only weakly sensitive to specific LP optimizer settings, since modern optimizers preserve query semantics; LP extraction failures correspond to compilation/syntax errors and are handled by iterative LLM regeneration before semantic validation starts.
>
> ---
>
> In summary, we sincerely appreciate the reviewers’ constructive feedback. **We provide a “Guideline for Reviewers” in the appendix, where all major revisions are highlighted with color to clearly indicate how each change corresponds to specific reviewer comments, enabling quick and convenient inspection**.

---

### Author Response · Authors · 2025-12-04

Dear SAC, AC, and PCs,

We sincerely appreciate your valuable time, insightful feedback, and constructive suggestions on our manuscript. Each comment played a crucial role in further refining and strengthening our work.

In response, we refined the theoretical exposition, experimental design, interpretability analysis, and overall presentation in our work.
All concerns have been addressed, and reviewers can follow the guidelines in **Appendix K** to locate the corresponding part in our manuscript.
Below, we provide a concise summary of the major revisions and clarifications throughout the updated version.

---

>### **Impact on Text-to-SQL Pipeline with HEROSQL Validator (Reviewer LazQ, Z8MB, and mSpA)**

We demonstrate that **HEROSQL can improve the end-to-end performance of Text-to-SQL pipelines** on the BIRD dataset with detailed results in Table 2, Section 5 of the revised manuscript.
With fine-grained error detection capabilities, HEROSQL can **provide feedback signals at the sub-SQL level**, enabling LLMs to correct prior mistakes more effectively.

---

>### **Generalization Ability on More Challenge or Ambiguous Queries (Reviewer Z8MB, mSpA, and 3mDL)**

**HEROSQL is a framework compatible with any LLM-based pretrained embedding model and any message-passing neural network**.
This model-agnostic design enables HEROSQL to seamlessly benefit from ongoing advances in scalable GNNs/LLMs, improving the generalization of HEROSQL.
As for more complex schema and SQL queries, HEROSQL has **sufficient generality to process any syntactically correct SQL statement**.
As long as a SQL statement is syntactically valid, HEROSQL performs semantic validation.

Additionally, we evaluate the performance of HEROSQL on the challenge dataset Spider 2.0 and the ambiguous dataset Ambrosia in Appendix H.1 and H.2.
Across all settings, **HEROSQL shows strong generalization ability and achieves SOTA performance** compared to baselines.

---

>### **Practical Considerations Regarding Latency  (Reviewer LazQ, Z8MB, and mSpA)**

We measured the inference latency and throughput for HEROSQL.
On RTX 3090 GPU, HEROSQL achieves an average end-to-end per-query latency of 11.7ms ($\approx$ 10ms), providing **competitive latency and throughput for interactive usage** and **meeting real-time industry validation constraints**.
Detailed results are presented in Appendix H.3.

---

>### **Clarification for the Fairness of Experimental Evaluation (Reviewer Z8MB, and 3mDL)**

In the experiments, we have combined domain-specific baselines with general LLM-based validators. Our setup closely **follows prior work**.
We clarified that **all baselines are implemented under rigorous and fully fair settings**, with detailed adaptations in Appendix E.1.
Further, ablation studies removing AST and LP demonstrate that HEROSQL’s gains arise from these components.

---

>### **Clarification for AST Perturbation (Reviewer Z8MB, and mSpA)**

Although the AST-driven augmented samples can sometimes preserve original semantics, we further apply execution-based filtering using the ground-truth answer.
Only those perturbed queries whose execution results differ from the ground truth are retained.
Additionally, on the training set, a mixture of LLM-generated samples and perturbed negatives is used to reduce overfitting to synthetic error patterns and improve overall robustness.

---

>### **Justification of Decoder-Only vs. Encoder-Only Baselines (Reviewer LazQ)**

We included additional experiments to justify that using decoder-only models, which have larger capacity and longer context windows, is preferable to encoder-only models.
More explanation and experiments are in Appendix B.

---

>### **Comparison with Commercial Models (Reviewer mSpA)**

In our experiments, we mainly use open-source backbones for privacy, cost, and efficiency reasons.
We also add experiments with strong commercial baselines and larger Qwen3 backbones.
Results in Table 14 of Appendix H.14 show that stronger LLMs produce better results, and scaling Qwen3 further enhances HEROSQL’s performance.
Moreover, HEROSQL consistently delivers added value regardless of which backbone it is applied to.

---

>### **Impact Of Number Of AST-Based Augmented Samples (Reviewer 3mDL)**

We added a detailed study varying the number of AST-based augmented samples by controlling negative-to-positive (N/P) ratio.
Results in Table 10 in Appendix G.1 show that maintaining a balanced N/P ratio of approximately 1:1 is an effective and robust strategy, which supports better generalization and more stable learning dynamics.

---

**We believe these clarifications, additional experiments, and extended analyses adequately address the concerns raised by the reviewers and more clearly highlight the technical soundness, methodological innovation, and empirical value of our work**. All comments have been addressed with the utmost care, and they have been invaluable in strengthening the revised manuscript.

---

### Meta-Review · Area_Chair_MK6r · 2026-01-05

**Summary:**

The paper proposes HeroSQL, a hierarchical representation framework for semantic validation in Text-to-SQL systems. The method integrates Global Intent (via Logical Plans - LP) and Local Details (via Abstract Syntax Trees - AST) using a Nested Message Passing Neural Network (NMPNN) to detect misalignments between natural language questions and generated SQL queries.

**Reviewer Concerns:**

The authors directly provided the requested latency data, reproducibility details, and justification for the embedding model choice, addressing all specific weaknesses cited. The concerns about scalability and robustness were addressed with the new Spider 2.0 and Ambrosia results. The conceptual concern about Logical Plans was effectively argued as a feature of validation (cross-view comparison) rather than a flaw. While the authors addressed the fairness of baselines and added generalization experiments (Spider 2.0), this reviewer's (3mDL) primary complaint was clarity/writing.

**Reviewer Scores:**

The reviewers initially recognized the novelty of the hierarchical representation and the strong empirical results on standard benchmarks. However, they raised significant concerns regarding the method's computational cost/latency , the justification for using decoder-only embeddings over encoder-only baselines , the robustness of the method to complex/noisy schemas , and the practical utility of the validator in end-to-end pipelines.

---

### Decision · Program_Chairs · 2026-01-26

Reject